# Incentivizing Combinatorial Bandit Exploration

Xinyan Hu[1], Dung Daniel Ngo[2], Aleksandrs Slivkins[3], and Zhiwei Steven Wu[4]

[1]University of California, Berkeley.* Email: xinyanhu@berkeley.edu
[2]University of Minnesota. Email: ngo00054@umn.edu
[3]Microsoft Research NYC. Email: slivkins@microsoft.com
[4]Carnegie Mellon University. Email: zstevenwu@cmu.edu

## Abstract

Consider a bandit algorithm that recommends actions to self-interested users in a recommendation system. The users are free to choose other actions and need to be incentivized to follow the algorithm's recommendations. While the users prefer to *exploit*, the algorithm can incentivize them to *explore* by leveraging the information collected from the previous users. All published work on this problem, known as *incentivized exploration*, focuses on small, unstructured action sets and mainly targets the case when the users' beliefs are independent across actions. However, realistic exploration problems often feature large, structured action sets and highly correlated beliefs. We focus on a paradigmatic exploration problem with structure: combinatorial semi-bandits. We prove that Thompson Sampling, when applied to combinatorial semi-bandits, is incentive-compatible when initialized with a sufficient number of samples of each arm (where this number is determined in advance by the Bayesian prior). Moreover, we design incentive-compatible algorithms for collecting the initial samples.

## 1 Introduction

We consider *incentivized exploration*: how to incentivize self-interested users to explore. A social planner interacts with self-interested users (henceforth, *agents*) and can make recommendations, but cannot enforce the agents to comply with these recommendations. The agents face uncertainty about the available alternatives. The social planner would like the agents to trade off *exploration* for the sake of acquiring new information and *exploitation*, making optimal near-term decisions based on the current information. The agents, on the other hand, prefer to *exploit*. However, the algorithm can incentivize them to *explore* by leveraging the information collected from the previous users. This problem has been studied since Kremer et al. (2014), see Slivkins (2019, Ch. 11) for an overview.

The basic model of incentivized exploration is very stylized and can be extended along two "dimension": more sophisticated economic models for agents' behavior and incentives, and more complex machine-learning models for actions' structures and rewards. All published work has only dealt with the former, whereas we pursue the latter. In particular, all published work focuses on small, unstructured action sets. Moreover, the case of *independent priors* – when the users' beliefs are independent across actions – is emphasized as the main, paradigmatic special case when specific performance guarantees are derived. However, realistic exploration problems often feature large sets with some known structure that connects actions to one another. A major recurring theme in the vast literature on multi-armed bandits is taking advantage of the available structure so as to enable the algorithm to cope with the large number of actions.

---

*Research done while X.Hu was an undergraduate student at Peking University and a (virtual) visiting student at Carnegie Mellon University.

36th Conference on Neural Information Processing Systems (NeurIPS 2022).

We focus on a paradigmatic, well-studied exploration problem with structured actions: *combinatorial semi-bandits*. Here, each arm is a subset of some ground set, whose elements are called *atoms*. In each round, the algorithm chooses an arm, and observes/collects reward for each atom in this arm. The reward for each atom is drawn independently from some fixed (but unknown) distribution specific to this atom. The set of feasible arms reflects the structure of the problem, *e.g.,* it can comprise all subsets of atoms of a given cardinality, or all edge-paths in a given graph. Since the number of arms $(K)$ can be exponential in the number of atoms $(d)$, the main theme is replacing the dependence on $K$ in regret bounds for "unstructured" $K$-armed bandits with a similar dependence on $d$.

We adopt a standard model for incentivized exploration from Kremer et al. (2014). The social planner is implemented as a bandit algorithm. Each round corresponds to a new agent which arrives and receives the arm chosen by the algorithm as a recommendation. Agents have Bayesian beliefs, independent across the atoms (but highly correlated across the arms). The algorithm must ensure that following its recommendation is in each agent's best interest, a condition called *Bayesian incentive-compatibility* (*BIC*). Each agent does not observe what happened with the previous agents, but the algorithm does. This information asymmetry is crucial for creating incentives.

**Our contributions.** We prove that Thompson Sampling is BIC when initialized with at least $n_{\text{TS}}$ samples of each atom, where $n_{\text{TS}}$ is determined by the prior and scales polynomially in the number of atoms $(d)$. Thompson Sampling (Thompson, 1933) is a well-known bandit algorithm with near-optimal regret bounds and good empirical performance. The initial samples can be provided by another BIC algorithm (more on this below), or procured exogenously, *e.g.,* bought with money.

Next, we consider the problem of *initial exploration*: essentially, design a BIC algorithm that samples each atom at least once. Such algorithms are interesting in their own right, and can be used to bootstrap Thompson Sampling, as per above. We present two such algorithms, which build on prior work (Mansour et al., 2020; Simchowitz and Slivkins, 2021) and extend it in non-trivial ways. The objective to be optimized is the sufficient number of rounds $T_0$, and particularly its dependence on $d$. To calibrate, prior work on incentivized exploration in multi-armed bandits with correlated priors does not provide any guarantees for a super-constant number of arms $(K)$, and is known to have $T_0 > \exp(\Omega(K))$ in some natural examples (Mansour et al., 2020). In contrast, our algorithms satisfy $T_0 \leq \exp(O(d))$ for a paradigmatic special case, and $T_0 \leq \exp(O(d^3))$ in general.

Finally, what if the prior is *not* independent across atoms? We focus on two arms with arbitrary correlation, a fundamental special case of incentivized exploration, and prove that our analysis of Thompson Sampling extends to the case. This result may be of independent interest.

**Discussion.** Like all prior work on incentivized exploration, we consider standard, yet idealized models for agents' economic behavior and the machine-learning problem being solved by the social planner. The modeling captures something essential about exploration and incentives in recommendation systems, but is not supposed to capture all the particularities of any specific application scenario. The goal of this paper is to bring more complexity into the machine-learning problem; advancing the economic model is beyond our scope.

We focus on establishing the BIC property and asymptotic guarantees in terms of the number of atoms, without attempting to optimize the dependence on the per-atom Bayesian priors. Our Thompson Sampling result has an encouraging practical implication: a standard, well-performing bandit algorithm plays well with users' incentives, provided a small (in theory) amount of initial data.

(Stylized) motivating examples of incentivized exploration in combinatorial semi-bandits include: recommending online content, *e.g.,* for news or entertainment (with atoms as *e.g.,* specific news articles); recommending complementary products, *e.g.,* a suit that consists of multiple items of clothing; recommending driving directions; see Appendix A for more details. In all examples, the social planner corresponds to the online platform issuing the respective recommendations. Such online platforms are often interested in maximizing users' happiness, rather than (or in addition to) the immediate revenue, as a way to ensure user engagement and long-term success.

**Related work.** Incentivized exploration, as defined in this paper, has been introduced in Kremer et al. (2014) and subsequently studied, *e.g.,* in Mansour et al. (2020, 2022); Immorlica et al. (2020); Bahar et al. (2016, 2019), along with some extensions. Most related is Sellke and Slivkins (2021), which obtains similar BIC results for the special case of multi-armed bandits with independent priors, both for Thompson Sampling and for initial exploration. A yet unpublished working paper of Simchowitz and Slivkins (2021) provides a BIC algorithm for initial exploration in reinforcement learning; we

build on this result in one of ours. Similar, but technically incomparable versions have been studied, *e.g.,* with time-discounted rewards (Bimpikis et al., 2018) and creating incentives via money (Frazier et al., 2014; Chen et al., 2018).

From the perspective of theoretical economics, incentivized exploration is related to the literature on information design (Kamenica, 2019; Bergemann and Morris, 2019): essentially, one round of incentivized exploration is an instance of Bayesian persuasion, a central model in this literature. Other "online" models of Bayesian persuasion have been studied (*e.g.,* Castiglioni et al., 2020; Zu et al., 2021), but are very different from ours in that the planner's private signal is drawn IID in each round (whereas in our model it is the algorithm's history); the problem has nothing to do with exploration, and is not even meaningful without incentives.

On the machine learning side, this paper is related to the work on combinatorial semi-bandits, starting from György et al. (2007), *e.g.,* (Chen et al., 2013; Kveton et al., 2015, 2014), and the work on Thompson Sampling, see Russo et al. (2018) for a survey. In particular, near-optimal Bayesian regret bounds have been derived in Russo and Van Roy (2014, 2016), and frequentist ones in (Agrawal and Goyal, 2017; Kaufmann et al., 2012). Thompson Sampling has been applied to combinatorial semi-bandits, (*e.g.,* Gopalan et al., 2014; Wen et al., 2015; Degenne and Perchet, 2016; Wang and Chen, 2018), with Bayesian regret bounds derived in Russo and Van Roy (2016).

## 2 Problem Formulation and Preliminaries

Our algorithm operates according to the standard protocol for *combinatorial semi-bandits*, with an ancillary incentive-compatibility constraint, standard in the literature on *incentivized exploration*.

**Combinatorial semi-bandits.** There are $T$ rounds, $d$ atoms and $K$ arms, where each arm is a subset of atoms. The set $\mathcal{A}$ of feasible arms is fixed and known. In each round $t$, each atom $\ell$ generates reward $r_\ell^{(t)} \in [0, 1]$. The algorithm chooses an arm $A^{(t)} \in \mathcal{A}$ and observes the reward of each atom in this arm (and nothing else). Algorithm's reward in this round is the total reward of these atoms.

Formally, we write $[T] := \{1, \ldots, T\}$ for the set of all rounds and $[d]$ for the set of all atoms, so that arms are subsets $A \subset [d]$. Let $\theta_\ell$ be the expected reward of atom $\ell \in [d]$, and let $\mu(A) = \sum_{\ell \in A} \theta_\ell$ be the expected reward of a given arm $A \subset [d]$. Note that $d$-armed bandits are a special case when the feasible arms are singleton sets $\{\ell\}$, $\ell \in [d]$.

**Stochastic rewards and Bayesian priors.** The reward of each atom $\ell \in [0, 1]$ is drawn independently in each round from a fixed distribution $\mathcal{D}_\ell$ specific to this atom. This distribution comes from a parametric family, parameterized by the expected reward $\theta_\ell$. The (realized) problem instance is therefore specified by the *mean reward vector* $\theta = (\theta_1, \ldots, \theta_d)$. Initially, each $\theta_\ell$ is drawn independently from a Bayesian prior $\mathcal{P}_\ell$ with support $\Theta \subset [0, 1]$. Put differently, the mean reward vector $\theta$ is drawn from the product prior $\mathcal{P} = \mathcal{P}_1 \times \cdots \times \mathcal{P}_d$.

**Incentive-compatibility.** The algorithm must ensure that in each round $t$, conditional on a particular arm $A = A^{(t)}$ being chosen, the expected reward of this arm is at least as good as that of any other arm. Formally, the algorithm is called *Bayesian incentive-compatible* (BIC) if for each round $t \in [T]$,

$$\mathbb{E}[\, \mu(A) - \mu(A') \mid A^{(t)} = A \,] \geq 0 \qquad \forall \text{ arms } A, A' \in \mathcal{A} \text{ with } \mathbb{P}[A^{(t)} = A] > 0. \tag{1}$$

This definition is based on the following stylized story . In each round $t$, a new user arrives to a recommendation system, observes the arm $A^{(t)}$ chosen by our algorithm, and interprets it as a recommendation. Then the user decides which arm to choose (not necessarily the arm recommended), and receives the corresponding reward. Accordingly, the user needs to be incentivized to follow the recommendation. We adopt a standard setup from economic theory (and the prior work on incentivized exploration): each user has the same prior $\mathcal{P}$, knows the algorithm, and wishes to maximize her expected reward. We posit that the user does not observe anything else before making her decision, other than the recommended arm. In particular, she does not observe anything about the previous rounds. Then, (1) ensures that she is (weakly) incentivized to follow her recommendation, assuming that the previous users followed theirs. We posit that under (1), the user does follow recommendations, and then reports the rewards of all atoms to the algorithm.

We emphasize that this story is not a part of our formal model (although it can be expressed as such if needed). In fact, the story can be extended to allow the algorithm to reveal an arbitrary "message"

to each user, but this additional power is useless: essentially, anything that can be achieved with arbitrary messages can also be achieved with arm recommendations. This can easily be proved as a version of Myerson's *direct revelation principle* from theoretical economics.

**Conventions.** Each atom $\ell$ satisfies $\mathbb{P}[\theta_\ell > 0] > 0$: else, its rewards are all 0, so it can be ignored.

No arm is contained in any the other arm. This is w.l.o.g. for Bernoulli rewards, and more generally if $\mathbb{P}[r_\ell^{(t)} = 0 \mid \theta_\ell > 0] > 0$: if $A \subset A'$ for arms $A, A'$ then $A$ cannot be chosen by any BIC algorithm.

W.l.o.g., order the atoms $\ell$ by their prior mean rewards $\theta_\ell^0 := \mathbb{E}[\theta_\ell]$, so that $\mathbb{E}[\theta_1^0] \geq \cdots \geq \mathbb{E}[\theta_d^0]$.

Let $A^* = \arg\max_{A \in \mathcal{A}} \mu(A)$ denote the best arm overall, with some fixed tie-breaking rule.

By a slight abuse of notation, each arm $A \subset [d]$ is sometimes identified with a binary vector $v \in \{0, 1\}^d$ such that $v_\ell = 1 \Leftrightarrow \ell \in A$, for each atom $\ell \in [d]$. In particular, we write $A_\ell = v_\ell$.

**Thompson Sampling** has a very simple definition, generic to many versions of multi-armed bandits. Let $\mathcal{F}_t$ denote the realized history (tuples of chosen actions and realized rewards of all atoms) up to and not including round $t$. Write $\mathbb{E}^{(t)}[\cdot] = \mathbb{E}[\cdot \mid \mathcal{F}_t]$ and $\mathbb{P}^{(t)}[\cdot] = \mathbb{P}[\cdot \mid \mathcal{F}_t]$ as a shorthand for posterior updates. Thompson Sampling in a given round $t$ draws an arm independently at random from distribution $p^{(t)}(A) = \mathbb{P}^{(t)}[A^* = A]$, $A \in \mathcal{A}$. If Thompson Sampling is started from some fixed round $t_0 > 1$, this is tantamount to starting the algorithm from round 1, but with prior $\mathcal{P}(\cdot \mid \mathcal{F}_{t_0})$ rather than $\mathcal{P}$. The algorithm is well-defined for an arbitrary prior $\mathcal{P}$.

While this paper is not concerned with computational issues, they are as follows. The posterior update $\mathcal{P}(\cdot \mid \mathcal{F}_t)$ can be performed for each atom $\ell$ separately: $\mathcal{P}_\ell(\cdot \mid \mathcal{F}_{t,\ell})$, where $\mathcal{F}_{t,\ell}$ is the corresponding history of samples from this atom. A standard implementation draws $\theta_\ell' \in [0, 1]$ independently from $\mathcal{P}_\ell(\cdot \mid \mathcal{F}_{t,\ell})$, for each atom $\ell$, then chooses the best arm according to these draws: $\arg\max_{A \in \mathcal{A}} \sum_{\ell \in A} \theta_\ell'$. The posterior updates $\mathcal{P}_\ell(\cdot \mid \mathcal{F}_{t,\ell})$ and the $\arg\max$ choice are not computationally efficient in general, and may require heuristics (this is a common situation for all variants of Thompson Sampling). A paradigmatic special case is Beta priors $\mathcal{P}_\ell$ and Bernoulli reward distributions $\mathcal{D}_\ell$, so that the posterior update $\mathcal{P}_\ell(\cdot \mid \mathcal{F}_{t,\ell})$ is another Beta prior.

**Composition of BIC algorithms.** We rely on a generic observation from Mansour et al. (2020, 2022) that the composition of two BIC algorithms is also BIC.

**Lemma 2.1.** *Let* ALG *be a BIC algorithm which stops after some round $T_0$. Let* ALG$'(H)$ *be another algorithm that initially inputs the history $H$ collected by* ALG, *and suppose it is BIC. Consider the composite algorithm:* ALG *followed by* ALG$'(H)$, *which stops at the time horizon $T$. If $T_0$ is determined before the composite algorithm starts, then this algorithm is BIC.*

# 3 Thompson Sampling is BIC

Our main result is that Thompson Sampling is BIC when initialized with at least $n_{\texttt{TS}}$ samples of each atom, where $n_{\texttt{TS}}$ is determined by the prior and scales polynomially in $d$, the number of atoms.

**Theorem 3.1.** *Let* ALG *be any BIC algorithm such that by some time $T_0 \leq T$ (determined by the prior) it almost surely collects at least $n_{\texttt{TS}} = C_{\texttt{TS}} \cdot d^2 \cdot \epsilon_{\texttt{TS}}^{-2} \cdot \log(\delta_{\texttt{TS}}^{-1})$ samples from each atom, where*

$$\epsilon_{\texttt{TS}} = \min_{A, A' \in \mathcal{A}} \mathbb{E}[(\mu(A) - \mu(A'))_+] \quad and \quad \delta_{\texttt{TS}} = \min_{A \in \mathcal{A}} \mathbb{P}[A^* = A], \tag{2}$$

*and $C_{\texttt{TS}}$ is a large enough absolute constant. Consider the composite algorithm which runs* ALG *for the first $T_0$ rounds, followed by Thompson sampling. This algorithm is BIC.*

Note that $T_0$ and $n_{\texttt{TS}}$ are "constants" once the prior is fixed, in the sense that they do not depend on the time horizon $T$, the mean reward vector $\theta$, or the rewards in the data collected by ALG.

**Remark 3.2.** For statistical guarantees, consider Bayesian regret, *i.e.*, regret in expectation over the Bayesian prior. Bayesian regret of the composite algorithm in Theorem 3.1 is at most $T_0$ plus Bayesian regret of Thompson Sampling. The latter is $O(\sqrt{dT \log d})$ for any prior (Russo and Van Roy, 2016). For an end-to-end result, we provide a suitable ALG in Section 4.1, with a specific $T_0$.

**Remark 3.3.** We can invoke Lemma 2.1 since $T_0$ is determined in advance. So, it suffices to show that each round $t$ of Thompson Sampling satisfies the BIC condition (1).

Let us clarify the dependence on $d$. Note that parameters $\epsilon_{\text{TS}}$ and $\delta_{\text{TS}}$ may depend on $d$ through the prior. To separate the dependence on $d$ from that on the prior, we posit that each per-atom prior $\mathcal{P}_\ell$ belongs to a fixed collection $\mathcal{C}$. We make mild non-degeneracy assumptions:[2]

$$\mathbb{P}\left[\mu(A') < \mathbb{E}[\mu(A)]\right] > 0 \qquad \text{for all arms } A \neq A'. \tag{3}$$

$$\mathbb{P}[\theta_\ell > \tau] > 0 \qquad \text{for all atoms } \ell \in [d] \text{ and some } \tau \in (0,1). \tag{4}$$

$$\mathbb{P}[\theta_\ell < x] > \text{poly}(1/x) \cdot \exp(-x^{-\alpha}) \qquad \text{for all atoms } \ell \in [d], x \in (0, 1/2) \text{ and some } \alpha \geq 0. \tag{5}$$

**Corollary 3.4.** *Suppose all priors $\mathcal{P}_\ell$ of atoms $\ell \in [d]$ belong to some fixed, finite collection $\mathcal{C}$ of priors and assumptions (3-5) are satisfied with some absolute constants $\alpha, \tau$. Then $n_{\text{TS}} = O_{\mathcal{C}}(d^{3+\alpha} \log d)$, where $O_{\mathcal{C}}$ hides the absolute constants and the dependency on $\mathcal{C}$.*

**Remark 3.5.** The initial data can also be provided to Thompson Sampling exogenously (rather than via a BIC algorithm ALG), *e.g.,* purchased with money. More formally, one would need to provide a collection of (arm, atoms' rewards) datapoints such that each atom is sampled at least $n_{\text{TS}}$ times.[3]

*Proof Sketch for Theorem 3.1 (full proof in Appendix B).* In order to establish the BIC condition in (1) for Thompson Sampling, we first observe that $\mathbb{P}[A^* = A]$ is a positive prior-dependent constant for all arms $A$, so it suffices to prove $\mathbb{E}\left[\mathbb{E}^{(t)}[\mu(A) - \mu(A')] \cdot \mathbf{1}_{\{A^*=A\}}\right] \geq 0$ for all $A, A'$.

Next, to show a lower bound on $\mathbb{E}\left[(\mu(A) - \mu(A')) \cdot \mathbf{1}_{\{A^*=A\}}\right]$, we will leverage the Harris inequality Theorem D.2, which says increasing functions of independent random variables are non-negatively correlated. Observe that the functions $(\mu(A) - \mu(A'))_+$ and $\mathbf{1}_{\{A^*=A\}}$ are co-monotone in each coordinate of $\theta$ (*i.e.,* either both increasing or both decreasing in a coordinate). Then, the mixed-monotonicity Harris inequality (see Theorem D.2) implies that:

$$\mathbb{E}\left[(\mu(A) - \mu(A')) \cdot \mathbf{1}_{\{A^*=A\}}\right] = \mathbb{E}\left[(\mu(A) - \mu(A'))_+ \cdot \mathbf{1}_{\{A^*=A\}}\right] \geq \epsilon_{\text{TS}} \cdot \delta_A \tag{6}$$

where $\delta_A = \mathbb{P}[A^* = A] \geq \delta_{\text{TS}}$.

To finish the proof, we show the expected absolute difference between $\mathbb{E}^{(t)}[\mu(A) - \mu(A')] \cdot \mathbf{1}_{\{A^*=A\}}$ and $(\mu(A) - \mu(A')) \cdot \mathbf{1}_{\{A^*=A\}}$ is upper bounded by $\epsilon_{\text{TS}} \cdot \delta_A$. By regrouping and using triangle inequality as well as $\left|x \cdot \mathbf{1}_{\{A^*=A\}}\right| = |x| \cdot \mathbf{1}_{\{A^*=A\}}$, we can upper bound this estimation error by sum of $\mathbb{E}\left[\left|\mathbb{E}^{(t)}\left[\mu(A)\right] - \mu(A)\right| \cdot \mathbf{1}_{\{A^*=A\}}\right]$ and $\mathbb{E}\left[\left|\mathbb{E}^{(t)}\left[\mu(A')\right] - \mu(A')\right| \cdot \mathbf{1}_{\{A^*=A\}}\right]$. Since the mean reward of each atom can be estimated by their empirical average, we can apply Bayesian Chernoff (see Lemma D.1) and observe that these two terms are $n_{\text{TS}}^{-1/2}$ times $O(1)$-sub-Gaussian random variables. By the sub-Gaussian tail bound (see Lemma D.4), we upper bound both terms by $O(n_{\text{TS}}^{-1/2} \delta_A \sqrt{\log(1/\delta_A)})$. We conclude by using our choices of $n_{\text{TS}}$ and observing that $\delta_{\text{TS}} \leq \delta_A$. $\square$

*Proof Sketch for Corollary 3.4 (full proof in Appendix B).* To derive the dependence of $n_{\text{TS}}$ on $d$, we investigate how the prior-dependent constants $\epsilon_{\text{TS}}$ and $\delta_{\text{TS}}$ depends on $d$. First, we can let $\epsilon_{\mathcal{C}}$ be a version of $\epsilon_{\text{TS}}$ where the min is taken over all ordered pairs of priors in $\mathcal{C}$. Since $\mathcal{C}$ is finite and satisfies the pairwise non-dominance assumption (3), $\epsilon_{\text{TS}} \geq \epsilon_{\mathcal{C}} > 0$.

By definition, $\delta_{\text{TS}} = \min_{A \in \mathcal{A}} \mathbb{P}[A^* = A]$ is the minimum probability that arm $A$ is the best arm overall. Fix an arm $A$. We observe that the event where arm $A$ is the best arm is more likely than the event where each atom in $A$ is larger than $\tau$, and all other atoms not in $A$ is smaller than $\tau/d$. Hence, we can lower bound $\mathbb{P}[A^* = A]$ by $\mathbb{E}\left[\mathbf{1}_{\{\forall \ell \in A, \theta_\ell \geq \tau\}} \cdot \mathbf{1}_{\{\forall x \notin A, \theta_x \leq \tau/d\}}\right]$. Since the prior $\mathcal{P}$ is independent across atoms, we can write the expression above as product of $\mathbb{E}\left[\mathbf{1}_{\{\forall \ell \in A, \theta_\ell \geq \tau\}}\right]$ and $\mathbb{E}\left[\mathbf{1}_{\{\forall x \notin A, \theta_x \leq \tau/d\}}\right]$. As the values $\{\theta_\ell\}_{\ell \in [d]}$ are independent and co-monotone in each in coordinate of $\theta$, repeated application of mixed-monotonicity Harris inequality (see Remark D.3) implies that:

$$\mathbb{P}[A^* = A] \geq \prod_{\ell \in A} \mathbb{E}\left[\mathbf{1}_{\{\theta_\ell \geq \tau\}}\right] \cdot \prod_{x \notin A} \mathbb{E}\left[\mathbf{1}_{\{\theta_x \leq \tau/d\}}\right] = \prod_{\ell \in A} \mathbb{P}[\theta_\ell \geq \tau] \cdot \prod_{x \notin A} \mathbb{P}[\theta_x \leq \tau/d]$$

$$\geq \prod_{\ell=1}^{d} \mathbb{P}[\theta_\ell \geq \tau] \, \mathbb{P}[\theta_\ell \leq \tau/d]$$

---

[2]For the special case of $d$-armed bandits, assumption (3) is necessary and sufficient for the respective arm $A$ to be *explorable*: chosen in some round by some BIC algorithm (Sellke and Slivkins, 2021).

[3]A subtlety: the number of samples of each arm should be known in advance. This is because otherwise Bayesian update on this data may become dependent on the data-collection algorithm.

By full support assumption (4), we define a prior-dependent constant $\rho_\tau = \min_{A \in \mathcal{A}} \mathbb{P}[\theta_\ell \geq \tau] > 0$. Then, by definition of $\rho_\tau$ and the non-degeneracy assumption (5), the expression above is lower bounded by $\rho_\tau^d \cdot \text{poly}(d^d/(\tau)^d) \cdot \exp(-d(\tau/d)^{-\alpha})$. Plugging this bound into $n_{TS}$, we obtain $n_{TS} = O_{\mathcal{C}}(d^{3+\alpha} \log d)$. □

## 3.1 The two-arm case with arbitrary correlation

What if the prior is *not* independent across atoms? Our analysis extends to the case of $K = 2$ arms $A, A'$ with arbitrary correlation between the atoms. In fact, we do not assume combinatorial semi-bandit structure, and instead focus on the fundamental special case of incentivized exploration: when one has two arms $A, A'$ with arbitrary joint prior on $(\mu(A), \mu(A'))$. [4]

**Theorem 3.6.** *The assertion in Theorem 3.1 holds for the case when one has two arms $A, A'$ and an arbitrary joint prior on $(\mu(A), \mu(A'))$.*

This result completes our understanding of incentivized exploration with two correlated arms: indeed, a necessary and sufficient condition (and the algorithm) are known for collecting the initial data (Mansour et al., 2020). A similar result for two *independent* arms is in (Sellke and Slivkins, 2021).

The analysis is very similar to that of Theorem 3.1, and omitted. Unfortunately, our technique does not appear to extend to a larger number of arbitrarily correlated arms.

# 4 BIC algorithms for initial exploration

We present two BIC algorithms for *initial exploration*, where the objective is to sample each atom at least once (*i.e.,* choose arms whose union is $[d]$) and complete in $N_0$ rounds for some $N_0$ determined by the prior. Such algorithms are interesting in their own right, and can be used to bootstrap Thompson Sampling as per Theorem 3.1. (To collect $n$ samples of each arm, repeat the algorithm $n$ times.) Both algorithms complete in the number of rounds that is exponential in $\text{poly}(d)$. The first algorithm completes in $\exp(O_{\mathcal{P}}(d))$ rounds, but is restricted to arms of the same size and Beta-Bernoulli priors. We obtain $\exp(O_{\mathcal{P}}(d^2))$ for arbitrary sets of arms. The second algorithm sidesteps the Beta-Bernoulli restriction, but completes in $\exp(O_{\mathcal{P}}(d^3))$ rounds.

## 4.1 Reduction to $K$-armed bandits

The first algorithm builds a substantial "super-structure" on top of the "hidden exploration" paradigm from Mansour et al. (2020). The latter paradigm is defined for $K$-armed bandits and is proved to work for arms with independent priors, ordered by their prior mean rewards. However, for combinatorial semi-bandits, the arms' priors are highly correlated. We provide a new interpretation of their analysis in terms of exploring a generic sequence of arms that satisfies a certain property (P). Our technical contribution is to construct a sequence of arms and prove that it satisfies Property (P). Note that it suffices to explore a sequence of arms which collectively cover all the atoms.

Throughout this subsection, we make the following assumptions:

$$\text{the prior } \mathcal{P}_\ell \text{ for each atom } \ell \text{ is a Beta distribution with parameters } (\alpha_\ell, \beta_\ell); \quad (7)$$

$$\text{the reward distributions } \mathcal{D}_\ell \text{ are Bernoulli distributions.} \quad (8)$$

This is a paradigmatic special case for Thompson Sampling (and Bayesian inference in general).

Let $\nu_\ell(n) = \alpha_\ell / (\alpha_\ell + \beta_\ell + n)$, $n \geq 0$ be the posterior mean reward of atom $\ell$ when conditioned on $n$ samples of this atom such that each of these samples returns reward 0.

Given any number $n \in \mathbb{N}$, let us define a sequence of $\kappa(n) \leq \infty$ arms $V_1^n, \ldots, V_{\kappa(n)}^n \in \mathcal{A}$. Let $V_1$ be a prior-best arm: any arm with the largest prior mean reward. The subsequent arms are defined inductively. Essentially, we pretend that each atom in each arm in the sequence so far has been sampled exactly $n$ times and received 0 each time it has been sampled. The next arm is defined as the posterior-best arm: an arm with a largest posterior reward after seing these samples. Formally, for each $i \geq 2$, we define arm $V_i^n$ given the previous arms $V_1^n, \ldots, V_{i-1}^n$. For each atom $\ell \in [d]$ define

---

[4]Equivalently, we have $d = 2$ atoms with an arbitrary joint prior on $(\theta_1, \theta_2)$, and the feasible arms are the two singleton arms $\{1\}$ and $\{2\}$.

$Z_i^n(\ell) = n$ if this atom is contained in one of the previous arms in the sequence, and set $Z_i^n(\ell) = 0$ otherwise. Then, define $V_i^n$ as the posterior-best arm if the posterior mean rewards for atoms $\ell$ are given by $\nu_\ell(Z_i^n(\ell))$. That is:

$$V_i^n \in \arg\max_{A \in \mathcal{A}} \sum_{\ell \in A} \nu_\ell(Z_i^n(\ell)). \tag{9}$$

The sequence stops when the arms therein cover all atoms at least once, and continues infinitely otherwise; this defines $\kappa(n)$. [5]

To state Property (P), we focus on this sequence for a particular, prior-dependent choice of $n$.

(P) There exist numbers $n_\mathcal{P} \in \mathbb{N}$ and $\tau_\mathcal{P}, \rho_\mathcal{P} \in (0, 1)$, determined by the prior $\mathcal{P}$, which satisfy the following. Focus on the sequence of arms $V_1, \ldots, V_\kappa$, where $\kappa = \kappa(n_\mathcal{P})$ and $V_i = V_i^{n_\mathcal{P}}$ for each $i \in [\kappa]$. Let $H_i^N$, $i \in [\kappa]$ be a dataset that consists of exactly $N \in \mathbb{N}$ samples of each arm $V_1, \ldots, V_i$, where each sample contains the reward for each atom in the respective arm, and $H_0^N$ is an empty dataset. Then

$$\mathbb{P}\left[X_i^N \geq \tau_\mathcal{P}\right] \geq \rho_\mathcal{P} \quad \forall i \in [\kappa] \text{ and } N \geq n_\mathcal{P}, \tag{10}$$

where the random variable $X_i^N$ is defined as

$$X_i^N = \min_{\text{arms } A \neq V_i} \mathbb{E}\left[\mu(V_i) - \mu(A) \mid H_{i-1}^N\right].$$

In intuition, any given arm $V_i$ can be the posterior best arm with a margin $\tau_\mathcal{P}$ and probability at least $\rho_\mathcal{P}$ after seeing at least $n_\mathcal{P}$ samples of the previous arms $V_1, \ldots, V_{i-1}$.

Given Property (P), prior work guarantees the following (without relying on assumptions (7-8)).

**Theorem 4.1** (Mansour et al. (2020)). *Assume Property (P) holds with constants $n_\mathcal{P}, \tau_\mathcal{P}, \rho_\mathcal{P}$ and $\kappa = \kappa(n_\mathcal{P})$. Then there exists a BIC algorithm which explores each arm $V_1, \ldots, V_\kappa$ at least $n_\mathcal{P}$ times and completes in $T_0$ rounds, where $T_0 = \kappa \cdot n_\mathcal{P} \cdot (1 + d) / (\tau_\mathcal{P} \cdot \rho_\mathcal{P})$.*

Next, we establish (P). First we state a result for a paradigmatic case when all arms have the same cardinality, then relax it in what follows (with a somewhat weaker guarantee).

**Theorem 4.2.** *Assume Beta-Bernoulli priors (7-8). Further, assume that*

$$\textit{The arms are all subsets of } [d] \textit{ of a given size } m; \tag{11}$$

*Then Property (P) holds with $\kappa = \kappa(n_\mathcal{P}) = \lceil d/m \rceil$ and*

$$n_\mathcal{P} = \lceil \beta_d / \alpha_d \rceil \cdot \max_{\ell \in [d]} \lceil \alpha_\ell \rceil \tag{12}$$

$$\tau_\mathcal{P} = \min_{\text{atoms } \ell \neq \ell' \in [d], \, n, n' \in \{0, n_\mathcal{P}\}} |\nu_\ell(n) - \nu_{\ell'}(n')|. \tag{13}$$

$$\rho_\mathcal{P} = (1 - \theta_1^0)^{d \cdot n_\mathcal{P}}, \tag{14}$$

*as long as $\tau_\mathcal{P}$ and $\rho_\mathcal{P}$ are strictly positive.*

*Proof Sketch.* For each arm $V_i$, $i \in [\kappa]$ we consider the event that the dataset $H_i^n$ from Property (P) contains the reward of $0$ for each samples of each atom. We take $n$ to be large enough so that this event makes all arms $V_1, \ldots, V_{i-1}$ look inferior to $V_i$, in terms of the posterior mean reward. The key is to lower-bound the probability of this event; a non-trivial step here requires Harris inequality. $\square$

We show that $T_0$, the requisite number of rounds, depends exponentially on the number of atoms $d$. To this end, we define a suitable parameterization of the priors. To handle $\tau_\mathcal{P}$ in Theorem 4.2, we posit a lower bound that depends on $d$, but this dependence is very mild.

---

[5]In Theorem 4.2 and Theorem 4.4, we upper-bound $\kappa(n)$ for some prior-dependent $n = n_\mathcal{P}$.

**Corollary 4.3.** *Assume Beta-Bernoulli priors (7-8) and that (11) holds. Fix some absolute constants $c_0 \in \mathbb{N}$ and $c, c' \in (0, 1)$. Suppose $\mathbb{E}[\theta_\ell] \leq c'$ for all atoms, and the priors satisfy the following non-degeneracy conditions:*

$$\max_{\ell, \ell' \in [d]} \lceil \beta_\ell / \alpha_\ell \rceil \cdot \lceil \alpha_{\ell'} \rceil \leq c_0,$$

$$\min_{\ell, \ell' \in [d], \; n, n' \in \{0, c_0\}} |\nu_\ell(n) - \nu_{\ell'}(n')| \geq \Omega(c^{-d})$$

*Then there exists a BIC algorithm which samples each atom at least once and completes in*

$$N_0 = O\left(c_0 \, d \cdot \Phi^d\right)$$

*rounds, where $\Phi = c \cdot (1 - c')^{-c_0}$ is a constant.*

Finally, we handle general feasible sets, *i.e.,* without assumption (11). The guarantee becomes slightly weaker, in that we have $d^2$ in the exponent rather than $d$.

**Theorem 4.4.** *Assume Beta-Bernoulli priors (7-8). Then Property (P) holds with $\kappa(n_{\mathcal{P}}) \leq d$ and*

$$n_{\mathcal{P}} = \lceil (\alpha_d + \beta_d)/\alpha_d \rceil \cdot \max_{\ell \in [d]} \lceil \alpha_\ell \rceil \cdot d \tag{15}$$

$$\tau_{\mathcal{P}} = \min_{A \neq A' \in \mathcal{A}, n \in \{0, n_{\mathcal{P}}\}^d} \left| \sum_{\ell \in A} \nu_\ell(n_\ell) - \sum_{\ell' \in A'} \nu_{\ell'}(n_{\ell'}) \right|. \tag{16}$$

$$\rho_{\mathcal{P}} = (1 - \theta_1^0)^{d \cdot n_{\mathcal{P}}}, \tag{17}$$

*as long as $\tau_{\mathcal{P}}$ and $\rho_{\mathcal{P}}$ are strictly positive.*

**Corollary 4.5.** *Assume Beta-Bernoulli priors (7-8). Fix some absolute constants $c_1 \in \mathbb{N}$ and $c_2, c_3 \in (0, 1)$. Suppose $\mathbb{E}\left[\theta_\ell\right] \leq c_3$ for all atoms, and the priors satisfy the following non-degeneracy conditions:*

$$\max_{\text{atoms } \ell, \ell' \in [d]} \lceil (\alpha_\ell + \beta_\ell)/\alpha_\ell \rceil \cdot \lceil \alpha_{\ell'} \rceil \leq c_1$$

$$\min_{A \neq A' \in \mathcal{A}, n \in \{0, n_{\mathcal{P}}\}^d} \left| \sum_{\ell \in A} \nu_\ell(n_\ell) - \sum_{\ell' \in A'} \nu_{\ell'}(n_{\ell'}) \right| \geq \Omega(c_2^{-d^2}). \tag{18}$$

*Then there exists a BIC algorithm which samples each atom at least once and completes in*

$$N_0 = O\left(c_1 \, d^3 \cdot \Phi^{d^2}\right)$$

*rounds, where $\Phi = c_2 \cdot (1 - c_3)^{-c_1}$ is a constant.*

In Appendix C.4, we provide some motivation for why (18) is a mild assumption. Our intuition is that "typically" $\tau_{\mathcal{P}}$ should be on the order of $e^{-O(d)}$, whereas (18) only requires it to be $\geq e^{-\Omega(d^2)}$.

## 4.2 Reduction to incentivized reinforcement learning

Our second algorithm builds on the Hidden Hallucination approach from Simchowitz and Slivkins (2021), which targets incentivized exploration for episodic reinforcement learning. We use this approach by "encoding" a problem instance of combinatorial semi-bandits as a tabular MDP, so that actions in the MDP correspond to atoms, and feasible trajectories correspond to feasible arms. Then we invoke a theorem in Simchowitz and Slivkins (2021) and "translate" this theorem back to combinatorial semi-bandits.

More specifically, consider a tabular MDP with deterministic transitions and unique initial state. Each action in this MDP correspond to some atom $\ell$; then the action's reward is drawn from the corresponding reward distribution $\mathcal{D}_\ell$. In general, only a subset of actions is feasible at a given state-stage pair of the MDP. Let $G$ be the transition graph in such MDP: it is a rooted directed graph such that the nodes of $G$ correspond to state-stage pairs in the MDP (the root node corresponding to the initial state and stage 0). Each edge $(u, v)$ in $G$ corresponds to some MDP action feasible at $u$, *i.e.,* to some atom. While different edges in $G$ can correspond to the same atom, we require that any rooted directed

path in $G$ cannot contain two edges that correspond to the same atom. Let $A_P$ be the subset of atoms that corresponds to a given rooted directed path $P$, and let $\mathcal{A}_G = \{\, A_P : \text{rooted directed paths in } G \,\}$ be the family of arms "encoded" by $G$. A family of arms $\mathcal{A}$ is called *MDP-encodable* if $\mathcal{A} = \mathcal{A}_G$ for some transition graph $G$ as defined above with $O(d^2)$ nodes.

Our result applies to all MDP-encodable feasible sets. In particular, the set of all subsets of exactly $m$ atoms, for some fixed $m \le d$, is MDP-encodable. To see this, consider an MDP with $m$ stages and $d$ states, where each state $\ell \in [d]$ corresponds to the largest atom already included in the arm, and actions feasible at a given stage $i$ and state $\ell$ correspond to all atoms larger than $\ell$.

Our result allows for arbitrary per-atom priors $\mathcal{P}_\ell$, subject to a minor non-degeneracy condition, and reward distributions $\mathcal{D}_\ell$ that are supported on the same countable set.

**Theorem 4.6.** *Consider a feasible arm set $\mathcal{A}$ that is MDP-encodable, as defined above. Suppose the per-atom priors $\mathcal{P}_\ell$ lie in some fixed, finite collection $\mathcal{C}$ such that any $\mathcal{P}_\ell \in \mathcal{C}$ satisfies $\mathbb{P}[\theta_\ell \le \epsilon] > 0$ for all $\epsilon > 0$ and $\mathbb{E}[\theta_\ell] > 0$. Further, suppose all reward distributions $\mathcal{D}_\ell$ that are supported on the same countable set. Fix parameter $\delta \in (0,1)$. There is a BIC algorithm such that with probability at least $1 - \delta$ each atom is sampled at least once. This algorithm completes in $N_0$ rounds, where $N_0 = \Phi_{\mathcal{C}}^{-d^3} \cdot O_{\mathcal{C}}\big(\operatorname{poly}(d) \cdot \log(\delta^{-1})\big)$ for some constant $\Phi_{\mathcal{C}} \in (0,1)$ determined by collection $\mathcal{C}$.*

**Remark 4.7.** While the guarantee in Theorem 4.6 holds with probability $1 - \delta$, rather than almost surely, it suffices to bootstrap Thompson Sampling in Theorem 3.1. To see this, let ALG be an algorithm that runs for $T_0 = N_0 \cdot n_{\texttt{TS}} + d \cdot n_{\texttt{TS}}$ rounds (with $n_{\texttt{TS}}$ from Theorem 3.1), and proceeds as follows: first it repeats the algorithm from Theorem 4.6 $n_{\texttt{TS}}$ times, and in the remaining rounds it deterministically plays an arm with the largest prior mean reward (this algorithm is BIC). Define the "success event" as one in which ALG samples each atom $\ge n_{\texttt{TS}}$ times in the first $N_0 \cdot N_{\texttt{TS}}$ rounds. Consider *another* algorithm, $\texttt{ALG}^*$, which runs for $T_0$ rounds, coincides with ALG on the first $N_0 \cdot N_{\texttt{TS}}$ rounds, and in the remaining rounds coincides with ALG on the success event, and otherwise plays some arms so as to sample each atom at least once (this algorithm is not necessarily BIC). Now, if Thompson Sampling is preceded by $\texttt{ALG}^*$, then the analysis in Theorem 3.1 guarantees that each round $t$ of Thompson Sampling satisfies the BIC property (1), and does so with a strictly positive prior-dependent constant on the right hand side of (1). Therefore, the same holds for ALG, since it coincides with $\texttt{ALG}^*$ w.h.p., if the failure probability $\delta$ in Theorem 4.6 is chosen small enough.

## Acknowledgments and Disclosure of Funding

ZSW, DN were supported in part by the NSF FAI Award #1939606, NSF SCC Award #1952085, a Google Faculty Research Award, a J.P. Morgan Faculty Award, a Facebook Research Award, and a Mozilla Research Grant. Any opinions, findings, conclusions, or recommendations expressed in this material are those of the authors and not necessarily reflect the views of the National Science Foundation and other funding agencies. We would like to thank Keegan Harris and Max Simchowitz for brief collaborations, and Mark Sellke for insightful conversations on incentivized exploration.

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
