# Appendices for Incentivizing Combinatorial Bandit Exploration

## A   Motivating examples

We spell out three examples of incentivized exploration that are specific to combinatorial semi-bandits. We frame each example as a recommendation system that recommends "actions" with a combinatorial structure. A user then can choose to follow such recommendation (i.e., pursue the action), or deviate from it. The BIC condition ensures the former, in which case the user reports a "reward" for each atom in the action. Thus:

- Recommending online content such as news/entertainment articles. Here, an "action" is a slate of articles. The platform recommends each user a slate of articles, and the user can choose to "follow the recommendation": look at each article in the slate. The feedback for each article is either generated automatically from the interaction (*e.g.,* dwell time or how far the user has scrolled) or entered explicitly (*e.g.,* "thumbs up" or "thumbs down").

- Recommending complementary products: *e.g.,* a three-piece suit or the contents of a child's pencil box. Thus, an "action" is a slate of products. A user who follows such a recommendation buys each item in the slate, and eventually reports its quality (*e.g.,* "thumbs up" or "thumbs down").

- Recommending driving directions. Here, an "action" consists of a route, which is a sequence of segments. A driver can choose to follow this route, in which case the travel time for each segment is reported (*e.g.,* automatically by an app).

Our framing differs from the "usual" framing for combinatorial semi-bandits, much like the framing for incentivized exploration differs from that for multi-armed bandits.

## B   BIC analysis for Thompson Sampling (proofs for Section 3)

This appendix provides the proofs for Section 3, the BIC analysis of Thompson Sampling. Specifically, we prove Theorem 3.1 (that Thompson Sampling is BIC when initialized with sufficiently many samples) and Corollary 3.4 (that the sufficient number of samples is polynomial in $d$).

**Proof of Theorem 3.1**   By definition, Thompson sampling is BIC at a particular round $t > T_0$ if and only if we have $\mathbb{E}[\mu(A) - \mu(A')|A^{(t)} = A] \geq 0$ for all $(i, j)$ such that $i \neq j$. This condition can be written as:

$$\mathbb{E}[\mu(A) - \mu(A')|A^{(t)} = A] = \frac{\mathbb{E}\left[\mathbb{E}^{(t)}[\mu(A) - \mu(A')] \, \mathbb{P}^{(t)}[A^{(t)} = A]\right]}{\mathbb{P}[A^{(t)} = A]}$$

$$= \frac{\mathbb{E}\left[\mathbb{E}^{(t)}[\mu(A) - \mu(A')] \, \mathbb{P}^{(t)}[A^* = A]\right]}{\mathbb{P}[A^* = A]}$$

(by definition of Thompson Sampling)

Observe that the denominator $\mathbb{P}[A^* = A]$ is a positive prior-dependent constant. Hence, we only need to bound the numerator to satisfy the BIC condition.

Fixing arms $A, A'$, we can rewrite the numerator as:

$$\mathbb{E}\left[\mathbb{E}^{(t)}[\mu(A) - \mu(A')] \, \mathbb{P}^{(t)}[A^* = A]\right] = \mathbb{E}\left[\mathbb{E}^{(t)}\left[\mathbb{E}^{(t)}[\mu(A) - \mu(A')] \cdot \mathbf{1}_{\{A^* = A\}}\right]\right]$$

$$= \mathbb{E}\left[\mathbb{E}^{(t)}[\mu(A) - \mu(A')] \cdot \mathbf{1}_{\{A^* = A\}}\right]$$

For Thompson sampling to be BIC, it suffices to show that $\mathbb{E}\left[\mathbb{E}^{(t)}[\mu(A) - \mu(A')]\mathbf{1}_{\{A^* = A\}}\right] \geq 0$.

We first prove our observation that the functions $(\mu(A) - \mu(A'))_+$ and $\mathbf{1}_{\{A^*=A\}}$ are co-monotone in each coordinate of $\theta$, which means they are both increasing in some coordinates while both decreasing in the other coordinates. Specifically for any $\ell$-th coordinate of $\theta$, they are both increasing in $\theta_\ell$ (given all other coordinates in $\theta$ stay the same) if $A_\ell$ (the $\ell$-th coordinate of $A$) equals to 1. Otherwise, if $A_\ell = 0$, they are both decreasing in $\theta_\ell$.

Given any $\theta$ and $\theta'$ having the same coordinates $\theta_x = \theta'_x$ ($x \in [d]$) except for the $\ell$-th coordinate, $\theta_\ell > \theta'_\ell$, and for any arm $A' \neq A$,

$$\langle \theta, A - A' \rangle - \langle \theta', A - A' \rangle \tag{B.1}$$

$$= \sum_{x=1}^{d} \theta_x (A_x - A'_x) - \sum_{x=1}^{d} \theta'_x (A_x - A'_x) \tag{B.2}$$

$$= (\theta_\ell - \theta'_\ell)(A_\ell - A'_\ell) \qquad \text{(other coordinates are the same except } \ell)$$

$$\begin{cases} \geq 0, & \text{if } A_\ell = 1 \\ \leq 0, & \text{if } A_\ell = 0 \end{cases} \qquad (\theta_\ell > \theta'_\ell \text{ and } A_\ell, A'_\ell \in \{0,1\})$$

Note that $\mathbf{1}_{\{A^*=A\}} = \mathbf{1}_{\{\mu(A) - \mu(A') \geq 0, \forall A' \neq A\}}$. So if $A_\ell = 1$, then $\mu(A) - \mu(A')$ are increasing in $\theta_\ell$ for all $A' \neq A$, especially for $A_k = A'$. Hence, $\mathbf{1}_{\{A^*=A\}}$ and $\mu(A) - \mu(A')$ are both increasing in $\theta_\ell$. Otherwise, if $A_\ell = 0$, they are both decreasing in $\theta_\ell$.

Hence, we can apply Remark D.3 to lower bound the expression above as follows:

$$\mathbb{E}[\mu(A) - \mu(A') \cdot \mathbf{1}_{\{A^*=A\}}] = \mathbb{E}[(\mu(A) - \mu(A'))_+ \cdot \mathbf{1}_{\{A^*=A\}}]$$

$$\geq \min_{A,A' \in \mathcal{A}} \mathbb{E}[(\mu(A) - \mu(A'))_+] \, \mathbb{P}[A^* = A]$$

$$= \epsilon_{\text{TS}} \delta_A$$

where $\delta_A = \mathbb{P}[A^* = A] \geq \delta_{\text{TS}}$.

To finish the proof, we need the following inequality to hold:

$$\mathbb{E}[|\mathbb{E}^{T_0}[\langle \theta, A - A' \rangle] \cdot \mathbf{1}_{\{A^*=A\}} - \langle \theta, A - A' \rangle \cdot \mathbf{1}_{\{A^*=A\}}|] \leq \epsilon_{\text{TS}} \delta_A \tag{B.3}$$

where $\delta_A = \mathbb{P}[A^* = A]$. Regrouping and using triangle inequality on the LHS of Equation (B.3), we have:

$$\mathbb{E}\left[|\mathbb{E}^{T_0}[\langle \theta, A - A' \rangle] \cdot \mathbf{1}_{\{A^*=A\}} - \langle \theta, A - A' \rangle \cdot \mathbf{1}_{\{A^*=A\}}|\right]$$

$$\leq \mathbb{E}\left[|\mathbb{E}^{T_0}[\langle \theta, A \rangle] - \langle \theta, A \rangle| \cdot \mathbf{1}_{\{A^*=A\}}\right] + \mathbb{E}\left[|\mathbb{E}^{T_0}[\langle \theta, A' \rangle] - \langle \theta, A' \rangle| \cdot \mathbf{1}_{\{A^*=A\}}\right] \tag{B.4}$$

The final step is to bound each individual summand in the inequality above. By the Bayesian Chernoff Bound (Lemma D.1), we have $\|\mathbb{E}^{T_0}[\theta] - \theta\|$ is a $(n_{\text{TS}}^{-1/2}\sqrt{d})$ times $O(1)$-sub-Gaussian random variable. Then, by Cauchy-Schwarz inequality, we have

$$|\mathbb{E}^{T_0}[\langle \theta, A \rangle] - \langle \theta, A \rangle| \leq \|\mathbb{E}^{T_0}[\theta] - \theta\| \cdot \|A\|$$

$$\leq \sqrt{d}\|\mathbb{E}^{T_0}[\theta] - \theta\|$$

Hence, $|\mathbb{E}^{T_0}[\langle \theta, A \rangle] - \langle \theta, A \rangle|$ and $|\mathbb{E}^{T_0}[\langle \theta, A' \rangle] - \langle \theta, A' \rangle|$ have magnitude at most as large as a $(n_{\text{TS}}^{-1/2} \cdot d)$ times $O(1)$-sub-Gaussian random variable. Then, we can apply Lemma D.4 to both terms in the inequality (B.4) above and upper bound it by at most $O\left(\delta_A \cdot n_{\text{TS}}^{-1/2} \cdot d\sqrt{\log\left(\delta_A^{-1}\right)}\right)$. Then, using our choice of $n_{\text{TS}}$ and $\delta_{\text{TS}} \leq \delta_A$, we arrive at the conclusion.

**Proof of Corollary 3.4**    Recall that by Theorem 3.1, we have $n_{\text{TS}} = C_{\text{TS}} \cdot d^2 \cdot \epsilon_{\text{TS}}^{-2} \cdot \log(\delta_{\text{TS}}^{-1})$. Let $\epsilon_{\mathcal{C}}$ be the version of $\epsilon_{\text{TS}}$ where the $\min$ is taken over all ordered pairs of priors in $\mathcal{C}$. Then we have $\epsilon_{\text{TS}} \geq \epsilon_{\mathcal{C}}$. Since $\mathcal{C}$ is finite and satisfy the pairwise non-dominance assumption, $\epsilon_{\mathcal{C}}$ is strictly positive.

By definition, $\delta_{\text{TS}} = \min_{A \in \mathcal{A}} \mathbb{P}[A^* = A]$. Fix an arm $A$. Then, we can decompose the probability of arm $A$ being the best arm as:

$$\mathbb{P}[A^* = A] = \mathbb{P}[\langle \theta, A - A' \rangle \geq 0, \forall A' \neq A]$$

$$= \mathbb{P}\left[\sum_{\ell \in A} \theta_\ell - \sum_{x \in A'} \theta_x \geq 0, \forall A' \neq A\right] \tag{B.5}$$

We observe that the event when $A$ is the best arm is more likely than the event when each atom in $A$ is larger than $\tau$, and all other atoms not in $A$ is smaller than $\tau/d$. Hence, we can lower bound the probability above as:

$$\mathbb{P}\left[\sum_{\ell \in A} \theta_\ell - \sum_{x \in A'} \theta_x \geq 0, \forall A' \neq A\right]$$
$$\geq \mathbb{P}\left[\forall \ell \in A, \theta_\ell \geq \tau \quad \text{and} \quad \forall x \notin A, \theta_x \leq \tau/d\right]$$
$$= \mathbb{P}\left[\forall \ell \in A, \theta_\ell \geq \tau\right] \cdot \mathbb{P}\left[\forall x \notin A, \theta_x \leq \tau/d\right] \qquad \text{(the prior is independent across atoms)}$$
$$= \mathbb{E}\left[\prod_{\ell \in A} \mathbf{1}_{\{\theta_\ell \geq \tau\}}\right] \cdot \mathbb{E}\left[\prod_{x \notin A} \mathbf{1}_{\{\theta_x \leq \tau/d\}}\right]$$

Observe that the values $\{\theta_\ell\}_{\ell \in [d]}$ are independent, and each function $\mathbf{1}_{\{\theta_\ell \geq \tau\}}$ (and $\mathbf{1}_{\{\theta_\ell \leq \tau/d\}}$) are co-monotone in each coordinate of $\theta$. Then, repeated application of mixed-monotone Harris inequality (see Remark D.3) implies that

$$\mathbb{P}[A^* = A] \geq \prod_{\ell \in A} \mathbb{E}[\mathbf{1}_{\{\theta_\ell \geq \tau\}}] \cdot \prod_{x \notin A} \mathbb{E}[\mathbf{1}_{\{\theta_x \leq \tau/d\}}] \qquad \text{(mixed-monotonicity Harris)}$$
$$= \prod_{\ell \in A} \mathbb{P}[\theta_\ell \geq \tau] \cdot \prod_{x \notin A} \mathbb{P}[\theta_x \leq \tau/d]$$
$$\geq \prod_{\ell=1}^{d} \mathbb{P}[\theta_\ell \geq \tau] \, \mathbb{P}[\theta_\ell \leq \tau/d]$$

By the full support assumption Equation (4), we define a prior-dependent constant $\rho_\tau = \min_{\ell \in [d]} \mathbb{P}[\theta_\ell \geq \tau] > 0$. Then, by definition of $\rho_\tau$ and the non-degeneracy assumption Equation (5), we have:

$$\prod_{\ell=1}^{d} \mathbb{P}[\theta_\ell \geq \tau] \, \mathbb{P}[\theta_\ell \leq \tau/d] \geq \prod_{\ell=1}^{d} \rho_\tau^d \cdot \mathrm{poly}(d/\tau) \cdot \exp(-(\tau/d)^{-\alpha})$$
$$\geq \rho_\tau^d \cdot \mathrm{poly}(d^d/(\tau)^d) \cdot \exp(-d(\tau/d)^{-\alpha})$$

Substituting this expressions and $\epsilon_{\mathrm{TS}}$ into $n_{\mathrm{TS}}$, we have $n_{\mathrm{TS}} = O_\mathcal{C}(d^{3+\alpha} \log d)$.

## C   Initial exploration: reduction to $K$-armed bandits (proofs for Section 4.1)

### C.1   Theorem 4.1: the approach from Mansour et al. (2020)

Recall that we build on an approach from Mansour et al. (2020), encapsulated in Theorem 4.1. Let us clarify how this theorem follows from the material in Mansour et al. (2020).

The algorithm from Mansour et al. (2020) is modified in two ways: it explores the arms in the order given by the sequence $V_1, \ldots, V_\kappa$, and the observed outcome from playing a given arm now includes the rewards for all atoms in this arm. Let us spell out the resulting algorithm, for completeness.

---

**Algorithm 1:** Hidden Exploration (modification of Algorithm 3 in Mansour et al. (2020))

**Parameters:** $L, N \in \mathbb{N}$

**1** For the first $N$ rounds, recommend arm $V_1$.

**2** Let $s_1 = \left( r_\ell^{(t)} : \ell \in V_1, t \in [N] \right)$ be the tuple of all observed per-atom rewards from arm $V_1$;

**3** **for** *each arm $V_i$ in increasing order of $i$* **do**

**4**     Let $A^* = \arg\max_{A \in \mathcal{A}} \mathbb{E}[\mu(A) \mid s_1, \ldots, s_{i-1}]$, breaking ties favoring smaller index;

**5**     From the set $P$ of the next $L \cdot N$ rounds, pick a set $Q$ of $N$ rounds uniformly at random;

**6**     Every agent $p \in P - Q$ is recommended arm $A^*$;

**7**     Every agent $p \in Q$ is recommended arm $V_i$;

**8**     Let $s_i^N$ be the tuple of all per-atom rewards from arm $V_i$ observed in rounds $t \in Q$;

**9** **end**

---

The analysis in Section 5.2 of Mansour et al. (2020) carries over seamlessly to combinatorial semi-bandits, and yields the following guarantee:

**Lemma C.1** (Mansour et al. (2020)). *Assume Property (P) holds with constants $n_{\mathcal{P}}, \tau_{\mathcal{P}}, \rho_{\mathcal{P}}$ and $\kappa = \kappa(n_{\mathcal{P}}) < \infty$. Then Algorithm 1 with parameters $N \geq n_{\mathcal{P}}$ and $L$ satisfying (C.1) is BIC:*

$$L \geq 1 + \frac{\mu_{\max}^0 - \mu_{\min}^0}{\tau_{\mathcal{P}} \cdot \rho_{\mathcal{P}}}, \tag{C.1}$$

*where $\mu_{\max}^0 = \max_{A \in \mathcal{A}} \mathbb{E}[\mu(A)]$ and $\mu_{\min}^0 = \min_{A \in \mathcal{A}} \mathbb{E}[\mu(A)]$.*

*Proof of Theorem 4.1.* It remains to interpret and simplify the quantities in Lemma C.1. According to Lemma C.1, Algorithm 1 is BIC with parameters $N \geq n_{\mathcal{P}}$ and $L$ satisfying Equation (C.1). It suffices to take $N = n_{\mathcal{P}}$. Since $\theta_\ell^0 \in [0, 1]$ for any $\ell \in [d]$, we have $0 \leq \mu_{\min}^0 \leq \mu_{\max}^0 \leq d$ and $0 \leq \mu_{\max}^0 - \mu_{\min}^0 \leq d$. Additionally, $\tau_{\mathcal{P}}, \rho_{\mathcal{P}} \in (0, 1)$. So

$$1 + \frac{\mu_{\max}^0 - \mu_{\min}^0}{\tau_{\mathcal{P}} \cdot \rho_{\mathcal{P}}} \leq 1 + \frac{d}{\tau_{\mathcal{P}} \cdot \rho_{\mathcal{P}}} \leq \frac{1+d}{\tau_{\mathcal{P}} \cdot \rho_{\mathcal{P}}}.$$

And thus it suffices to take $L = \frac{1+d}{\tau_{\mathcal{P}} \cdot \rho_{\mathcal{P}}}$. Then we have the total number of rounds $T_0 = \kappa \cdot N \cdot L = \kappa \cdot n_{\mathcal{P}} \cdot (1+d) / (\tau_{\mathcal{P}} \cdot \rho_{\mathcal{P}})$. $\square$

## C.2 Restricted family of arms: Proof of Theorem 4.2

Firstly, according to Assumption (11) and the reward support $\Theta \subset [0, 1]$, we observe that:

$$\text{the prior/posterior-best arm contains the } m \text{ prior/posterior-best atoms;} \tag{C.2}$$
$$\text{the second prior/posterior-best arm contains the } m - 1 \text{ prior/posterior-best atoms.} \tag{C.3}$$

Then, according to our choice of $n_{\mathcal{P}}$ (12), we will prove that $\kappa = \kappa(n_{\mathcal{P}})$ is finite (i.e. our arm sequence will contain all atoms at least once). Actually, we will prove $\kappa = \lceil d/m \rceil$ by proving the following Claim C.2.

**Claim C.2.** *Assume Beta-Bernoulli priors (7-8), all arms have a fixed size (11), and $n_{\mathcal{P}}$ satisfies (12). Then the arm sequence $V_1, V_2, \cdots$, where $V_i = V_i^{n_{\mathcal{P}}}$, have the following properties:*

$$V_i = \{ (i-1)m + \ell : \ell \in [m] \}, \quad i \in [\lceil d/m \rceil - 1]; \tag{C.4}$$
$$V_i \supset \{ (i-1)m + \ell : \ell \in [m], (i-1)m + \ell \leq d \}, \quad i = \lceil d/m \rceil. \tag{C.5}$$

*And thus $\kappa(n_{\mathcal{P}}) = \lceil d/m \rceil$.*

*Proof.* We will prove by induction on phase $i$. For phase $i = 1$, $V_1$ is the prior-best arm. According to the observation (C.2), $V_1$ contains the largest $m$ prior-best atoms, which is $[m]$.

Suppose the induction hypothesis is true for all phases up to some phase $i \in [\lceil d/m \rceil - 1]$. Denote $B_i$ as a subset of atoms having been contained at least once in the first $i$ arms and $\bar{B}_i$ as the complement subset of atoms. Then

$$B_i = \bigcup_{j \in [i]} V_j = [im] \text{ and } \bar{B}_i = [d] - [im].$$

Recall the definition of $Z_\ell^{n_{\mathcal{P}}}$ and $\nu_\ell(n)$, we have for each atom $\ell \in B_i$ and $\ell' \in \bar{B}_i$:

$$Z_\ell^{n_{\mathcal{P}}} = n_{\mathcal{P}} \text{ and } \nu_\ell(Z_\ell^{n_{\mathcal{P}}}) = \nu_\ell(n_{\mathcal{P}}) = \alpha_\ell / (\alpha_\ell + \beta_\ell + n_{\mathcal{P}});$$

$$Z_{\ell'}^{n_{\mathcal{P}}} = 0 \text{ and } \nu_{\ell'}(Z_{\ell'}^{n_{\mathcal{P}}}) = \nu_{\ell'}(0) = \alpha_{\ell'} / (\alpha_{\ell'} + \beta_{\ell'}) = \theta_{\ell'}^0.$$

Since $\theta_1^0 \geq, \ldots, \geq \theta_d^0$, we have:

$$\nu_{\ell'}(Z_{\ell'}^{n_{\mathcal{P}}}) \text{ decreases in } \ell' \in \bar{B}_i. \tag{C.6}$$

By definition of $n_{\mathcal{P}}$ and $\theta_{\ell'}^0 \geq \theta_d^0$, we have:

$$\alpha_\ell / (\alpha_\ell + \beta_\ell + n_{\mathcal{P}}) < \alpha_d / (\alpha_d + \beta_d) \leq \alpha_{\ell'} / (\alpha_{\ell'} + \beta_{\ell'}) = \theta_{\ell'}^0.$$

Thus:

$$\nu_\ell(Z_\ell^{n_\mathcal{P}}) < \nu_{\ell'}(Z_{\ell'}^{n_\mathcal{P}}), \forall \ell \in B_i, \ell' \in \bar{B}_i. \tag{C.7}$$

Combining (C.6)-(C.7) and according to the observation (C.2), we have $V_{i+1}$ for phase $i+1$. If $i \in [\lceil d/m \rceil - 2]$, we have $|\bar{B}_i| = d - im \geq m + 1$. Thus $im + 1 < \cdots < (i+1)m \leq d - 1$ and $V_{i+1} = \{im + \ell : \ell \in [m]\}$. Otherwise for $i = \lceil d/m \rceil - 1$, we have $1 \leq |\bar{B}_i| \leq m$. Thus $V_{i+1} \supset \bar{B}_i = \{im + \ell : \ell \in [m] \text{ and } im + \ell \leq d\}$. Thus, the induction hypothesis is true for phase $i + 1$ and we complete the induction proof. And since $V_1, \ldots, V_{\lceil d/m \rceil}$ contain all atoms, we have $\kappa(n_\mathcal{P}) = \lceil d/m \rceil$. $\qquad\square$

Secondly, we define an event and give a lower bound of the probability of this event. Given any $n_\mathcal{P}, N \in \mathbb{N}$ ($n_\mathcal{P} \leq N$) and $H_i^N, \forall i \in [\kappa]$, define an event $\mathcal{E}_i$ for each $i \in [\kappa]$ saying that the first $n_\mathcal{P}$ reward samples of each atom in $\bigcup_{j \in [i]} V_j$ are 0. Formally,

$$\mathcal{E}_i = \left\{ r_\ell^{(t)} = 0, \forall \ell \in \bigcup_{j \in [i]} V_j, t \in [n_\mathcal{P}] \right\}, \forall i \in [\kappa]. \tag{C.8}$$

where we abuse the notation of $r_\ell^{(t)}$ as the $t$-th round that atom $\ell$ is being contained. Since $H_0^N$ is an empty data set, we define $\mathcal{E}_0$ is a full event, which gives no information wherever it applies.

Then, according to our choice of $\rho_\mathcal{P}$ (14), we will lower bound the probability of the event defined above in the following claim.

**Claim C.3.** *Assume Beta-Bernoulli priors (7-8) and $\rho_\mathcal{P}$ satisfies (14). Then for any given $n_\mathcal{P} \leq N$, with the definition of $\mathcal{E}_i, \forall i \in [\kappa]$ (C.8), we have:*

$$\mathbb{P}[\mathcal{E}_i] \geq \rho_\mathcal{P}, \forall i \in [\kappa]. \tag{C.9}$$

*Proof.* Firstly, by the prior and reward independence among each atoms:

$$\mathbb{P}[\mathcal{E}_i] = \mathbb{P}\left[ r_\ell^{(t)} = 0, \forall \ell \in \bigcup_{j \in [i]} V_j, t \in [n_\mathcal{P}] \right]$$

$$= \prod_{\ell \in \bigcup_{j \in [i]} V_j} \mathbb{P}\left[ r_\ell^{(t)} = 0, \forall t \in [n_\mathcal{P}] \right]$$

Secondly, for each given atom $\ell$, by the independence among realized rewards conditioned on the mean reward drawn from the prior and iteratively using Harris Inequality:

$$\mathbb{P}\left[ r_\ell^{(t)} = 0, \forall t \in [n_\mathcal{P}] \right] = \mathbb{E}_{\theta_\ell}\left[ \mathbb{P}_{r_\ell^{(t)}}\left[ r_\ell^{(t)} = 0, \forall t \in [n_\mathcal{P}] \right] \mid \theta_\ell \right]$$

$$= \mathbb{E}_{\theta_\ell}\left[ \prod_{t \in [n_\mathcal{P}]} \mathbb{P}_{r_\ell^{(t)}}\left[ r_\ell^{(t)} = 0 \mid \theta_\ell \right] \right] \qquad \text{(conditional independence)}$$

$$= \mathbb{E}_{\theta_\ell}\left[ \prod_{t \in [n_\mathcal{P}]} (1 - \theta_\ell) \right] \qquad \text{(Bernoulli rewards (8))}$$

$$\geq \prod_{t \in [n_\mathcal{P}]} \mathbb{E}_{\theta_\ell}\left[ (1 - \theta_\ell) \right] \qquad \text{(Harris inequality)}$$

$$= (1 - \theta_\ell^0)^{n_\mathcal{P}}$$

Combining both and recall that $\theta_1^0 \geq \cdots \geq \theta_d^0$:

$$\mathbb{P}[\mathcal{E}_i] \geq \prod_{\ell \in \bigcup_{j \in [i]} V_j} (1 - \theta_\ell^0)^{n_\mathcal{P}} \geq \prod_{\ell \in [d]} (1 - \theta_\ell^0)^{n_\mathcal{P}} \geq (1 - \theta_1^0)^{d n_\mathcal{P}} = \rho_\mathcal{P}.$$

$\qquad\square$

Note that this second part analysis does not rely on (11) and we will reuse that part for the proof of general feasible arm set case in Appendix C.3.

Thirdly, according to our choice of $\tau_{\mathcal{P}}$ (13), we will prove the following claim, which says the expectation $X_i^N$ conditioned on the event $\mathcal{E}_i$ almost surely $\geq \tau_{\mathcal{P}}$ for any phase $i \in [\kappa]$ and any $N \geq n_{\mathcal{P}}$.

**Claim C.4.** *Assume Beta-Bernoulli priors (7-8), all arms have a fixed size (11) and $\tau_{\mathcal{P}}$ satisfies (13). Then for any given $n_{\mathcal{P}}$, we have:*

$$\mathbb{P}\left[X_i^N \geq \tau_{\mathcal{P}} \mid \mathcal{E}_{i-1}\right] = 1, \forall i \in [\kappa], N \geq n_{\mathcal{P}}. \tag{C.10}$$

*Proof.* For each phase $i \in [\kappa]$, let $A_i$ is the second prior/posterior-best arm conditioned on $H_{i-1}$ and $\mathcal{E}_{i-1}$. According to observation (C.2)-(C.3) and the definition of $\tau_{\mathcal{P}}$:

$$\min_{\text{arm } A \neq V_i} \mathbb{E}\left[\mu(V_i) - \mu(A) \mid H_{i-1}^N, \mathcal{E}_{i-1}\right] = \mathbb{E}\left[\mu(V_i) - \mu(A_i) \mid H_{i-1}^N, \mathcal{E}_{i-1}\right]$$

$$= \sum_{\ell \in V_i} \nu_\ell(Z_\ell^{n_{\mathcal{P}}}) - \sum_{\ell' \in A_i} \nu_{\ell'}(Z_{\ell'}^{n_{\mathcal{P}}})$$

$$= \min_{\ell \in V_i} \nu_\ell(Z_\ell^{n_{\mathcal{P}}}) - \min_{\ell' \in A_i} \nu_\ell(Z_{\ell'}^{n_{\mathcal{P}}})$$

$$\geq \min_{\ell, \ell' \in [d], n, n' \in \{0, n_{\mathcal{P}}\}} |\nu_\ell(n) - \nu_{\ell'}(n')|$$

$$= \tau_{\mathcal{P}}.$$

Thus we have Equation (C.10). $\qquad\qquad\square$

At last, combining the claims above, we have for each $i \in [\kappa]$:

$$\mathbb{P}\left[X_i^N \geq \tau_{\mathcal{P}}\right] \geq \mathbb{P}\left[\mathcal{E}_{i-1}\right] \cdot \mathbb{P}\left[X_i^N \geq \tau_{\mathcal{P}} \mid \mathcal{E}_{i-1}\right]$$

$$\geq \rho_{\mathcal{P}} \cdot \mathbb{P}\left[X_i^N \geq \tau_{\mathcal{P}} \mid \mathcal{E}_{i-1}\right] \qquad\qquad \text{(Claim C.4)}$$

$$= \rho_{\mathcal{P}} \cdot 1 \qquad\qquad\qquad\qquad\qquad \text{(Claim C.3)}$$

$$= \rho_{\mathcal{P}},$$

which implies $(P)$.

## C.3  Arbitrary family of arms: Proof of Theorem 4.4

We prove Theorem 4.4, reusing much of the proof of Theorem 4.2. While the parameters in Theorem 4.4 give a weaker bound on the number of rounds, the proof becomes more intuitive.

Firstly, we prove $\kappa$ is finite in this following claim.

**Claim C.5.** *Assume Beta-Bernoulli priors (7-8) and $n_{\mathcal{P}}$ satisfies (15). Then $\kappa(n_{\mathcal{P}}) \leq d$.*

*Proof.* Denote the explored atom set up to phase $i$ as $B_i = \bigcup_{j \in [i]} V_j$ and the unexplored atom set as $\bar{B}_i = [d] - B_i$. Denote $B_0 = \emptyset$ and $\bar{B}_0 = [d]$. Fixed $i \geq 0$. Recall the definition of $Z_\ell^{n_{\mathcal{P}}}$ and $\nu_\ell(n)$, we have for each atom $\ell \in B_i$ and $\ell' \in \bar{B}_i$:

$$Z_\ell^{n_{\mathcal{P}}} = n_{\mathcal{P}} \text{ and } \nu_\ell(Z_\ell^{n_{\mathcal{P}}}) = \nu_\ell(n_{\mathcal{P}}) = \alpha_\ell/(\alpha_\ell + \beta_\ell + n_{\mathcal{P}});$$

$$Z_{\ell'}^{n_{\mathcal{P}}} = 0 \text{ and } \nu_{\ell'}(Z_{\ell'}^{n_{\mathcal{P}}}) = \nu_{\ell'}(0) = \alpha_{\ell'}/(\alpha_{\ell'} + \beta_{\ell'}).$$

By definition of $n_{\mathcal{P}}$ and $\theta_{\ell'}^0 \geq \theta_d^0$, we have:

$$\nu_{\ell'}(Z_{\ell'}^{n_{\mathcal{P}}}) = \alpha_\ell/(\alpha_\ell + \beta_\ell + n_{\mathcal{P}}) < \frac{1}{d}\alpha_d/(\alpha_d + \beta_d) \leq \frac{1}{d}\alpha_{\ell'}/(\alpha_{\ell'} + \beta_{\ell'}) = \frac{1}{d}\nu_{\ell'}(Z_{\ell'}^{n_{\mathcal{P}}}).$$

Thus:

$$\sum_{\ell \in B_i} \nu_\ell(Z_\ell^{n_{\mathcal{P}}}) < d \cdot \frac{1}{d}\nu_{\ell'}(Z_{\ell'}^{n_{\mathcal{P}}}) = \nu_{\ell'}(Z_{\ell'}^{n_{\mathcal{P}}}), \forall \ell' \in \bar{B}_i. \tag{C.11}$$

Then according to the definition of $V_{i+1}$, we know $V_{i+1}$ contains at least one atom $\ell' \in \bar{B}_i$. So the number of uncovered atoms, i.e. $|\bar{B}_i|$, decreases at least 1 after each phase. Thus it takes at most $d$ phases to cover all atoms, which implies $\kappa \leq d$. $\qquad\square$

Secondly, we reuse the definition of $\mathcal{E}_i$ (C.8) and Claim C.3 to give a lower bound of $\mathbb{P}[\mathcal{E}_i]$, since this part in Appendix C.2 don't rely on Assumption (11).

Thirdly, according to our definition of $\tau_{\mathcal{P}}$ (16), we have the following claim similar to Claim C.4.

**Claim C.6.** *Assume Beta-Bernoulli priors (7-8) and $\tau_{\mathcal{P}}$ satisfies (16). Then for any given $n_{\mathcal{P}}$, we have:*

$$\mathbb{P}\left[X_i^N \geq \tau_{\mathcal{P}} \mid \mathcal{E}_{i-1}\right] = 1, \forall i \in [\kappa], N \geq n_{\mathcal{P}}. \tag{C.12}$$

*Proof.* For each phase $i \in [\kappa]$, we have:

$$\min_{\text{arm } A \neq V_i} \mathbb{E}\left[\mu(V_i) - \mu(A) \mid H_{i-1}^N, \mathcal{E}_{i-1}\right] \geq \min_{A \neq A' \in \mathcal{A}} \mathbb{E}\left[\mu(A) - \mu(A') \mid H_{i-1}^N, \mathcal{E}_{i-1}\right]$$

$$= \min_{A \neq A' \in \mathcal{A}, n, n' \in \{0, n_{\mathcal{P}}\}^d} \left| \sum_{\ell \in A} \nu_\ell(n) - \sum_{\ell' \in A'} \nu_\ell(n') \right|$$

$$= \tau_{\mathcal{P}}.$$

Then we have Equation (C.12). □

At last, similar to the last step in Appendix C.2, combining Claim C.5, Claim C.3 and Claim C.6, we have for each $i \in [\kappa]$:

$$\begin{aligned}
\mathbb{P}\left[X_i^N \geq \tau_{\mathcal{P}}\right] &\geq \mathbb{P}[\mathcal{E}_{i-1}] \cdot \mathbb{P}\left[X_i^N \geq \tau_{\mathcal{P}} \mid \mathcal{E}_{i-1}\right] \\
&\geq \rho_{\mathcal{P}} \cdot \mathbb{P}\left[X_i^N \geq \tau_{\mathcal{P}} \mid \mathcal{E}_{i-1}\right] &\text{(Claim C.6)} \\
&= \rho_{\mathcal{P}} \cdot 1 &\text{(Claim C.3)} \\
&= \rho_{\mathcal{P}},
\end{aligned}$$

which implies $(P)$.

## C.4 Motivation for assumption (18)

Let us provide some motivation for why (18) is a mild assumption.

Fix a vector $n \in \mathbb{N}^d$ and define

$$\tau_{\mathcal{P}}(n) = \min_{A \neq A' \in \mathcal{A}} \left| \sum_{\ell \in A} \nu_\ell(n_\ell) - \sum_{\ell' \in A'} \nu_{\ell'}(n_{\ell'}) \right|. \tag{C.13}$$

Our intuition is as follows: $\tau_{\mathcal{P}}(n)$ is defined as the smallest difference between $e^{O(d)}$ numbers in $[-d, d]$, so typical situation should be that $\tau_{\mathcal{P}}$ is on the order of $e^{-O(d)}$, whereas our assumption only requires it to be larger than $e^{-\Omega(d^2)}$.

We make this intuition precise, in a sense defined below. We argue that $\tau_{\mathcal{P}}(n)$ is "not too small" for "all but a few" problem instances. More formally, we define a distribution over problem instances such that $\tau_{\mathcal{P}}(n) \geq \Omega(c_2^{-d^2})$ with very high probability. For instance, we can make it hold with probability at least $1 - \delta/2^d$ for some small $\delta > 0$.

(However, we do not construct one distribution that works for all relevant vectors $n$ at once, although we suspect that our technique, based on Esseen inequality, might be extended there.)

So, let us construct the desired distribution over problem instances. We fix $d$ and the set of feasible arms, and we only vary the per-atom priors. Recall that the prior $\mathcal{P}_\ell$ for a given atom $\ell \in [d]$ is specified by a pair of numbers, $(\alpha_\ell, \beta_\ell)$. Further, recall that $\nu_\ell(n_\ell) = \alpha_\ell / (\alpha_\ell + \beta_\ell + n_\ell), n_\ell \in \mathbb{N}$. We require that $\nu_\ell(n_\ell)$ is distributed uniformly on some interval.

**Lemma C.7.** *Fix vector $n \in \mathbb{N}^d$. Suppose for each atom $\ell \in [d]$, the pair $(\alpha_\ell, \beta_\ell)$ is drawn independently from some distribution such that $\nu_\ell(n_\ell)$ is uniformly distributed in some interval $[a_\ell, b_\ell]$. Fix $\delta \in (0, 1)$. Then it holds that*

$$\mathbb{P}\left[\tau_{\mathcal{P}}(n) < \frac{\delta}{2 \cdot 8^d}\right] \leq \frac{\delta}{2^d}. \tag{C.14}$$

**Remark C.8.** One way to ensure that $\nu_\ell(n_\ell)$ is uniformly distributed is as follows. Fix atom $\ell$, and parameters $\beta_\ell$ and $n_\ell$. Let $\nu_\ell = \nu_\ell(n_\ell)$. Note that

$$\alpha_\ell = \Phi(\nu_\ell) := \frac{\nu_\ell \cdot (\beta_\ell + n_\ell)}{1 - \nu_\ell}. \tag{C.15}$$

Now, just let $\alpha_\ell$ be distributed as $\Phi(Y)$, where $Y$ is uniform on $[a_\ell, b_\ell]$ interval, where $0 \leq a_\ell \leq \alpha_\ell \leq b_\ell \leq 1$. Observe that by change-of-variable, when $\alpha_\ell$ is distributed with $\Phi(Y)$, then $\nu_\ell$ is distributed uniformly on $[a_\ell, b_\ell]$.

To prove Lemma C.7, we invoke the following tool from *anti-concentration*.

**Theorem C.9** (Esseen inequality)**.** *Let $Y$ be a random variable. Consider its* characteristic function*,*

$$\psi_Y(\lambda) = \mathbb{E}\left[e^{i\lambda Y}\right], \quad \lambda \in \mathbb{R}.$$

*Then for any $x > 0$ it holds that*

$$Q_Y(x) := \sup_{y \in \mathbb{R}} \mathbb{P}\left[|Y - y| \leq x\right] \leq x \int_{-2\pi/x}^{2\pi/x} |\psi_Y(\lambda)| \; d\lambda.$$

*Proof of Lemma C.7.* Let $\nu_\ell = \nu_\ell(n_\ell)$ for each atom $\ell \in [d]$. Focus on

$$X := \sum_{\ell \in A} \nu_\ell - \sum_{\ell' \in A'} \nu_{\ell'}.$$

We treat $X$ as a random variable, under the distribution over the $(\alpha_\ell, \beta_\ell)$ pairs. Without loss of generality, from here on assume that arms $A$ and $A'$ are disjoint subsets of atoms.

We will use Theorem C.9 to prove that $Q_X(x) \leq 2x$ for any $x > 0$.

Since $X$ is a sum of independent random variables $\pm \nu_\ell$, $\ell \in A \cup A'$, the characteristic function of $X$ is the product of the respective characteristic functions

$$\psi_X(\lambda) = \prod_{\ell \in A \cup A'} \psi_\ell(\lambda),$$

where $\psi_\ell(\lambda) = \psi_{\nu_\ell}(\lambda)$ for $\ell \in A$, and $\psi_\ell(\lambda) = \psi_{-\nu_\ell}(\lambda)$ for $\ell \in A'$.

From here on, fix some atom $\ell \in A$. Since $|\psi_Y(\lambda)| \leq 1$ for any random variable $Y$, it follows that

$$|\psi_X(\lambda)| \leq |\psi_{\nu_\ell}(\lambda)|.$$

For the rest of the proof we focus on the characteristic function for $\nu_\ell$, $\psi(\cdot) := \psi_{\nu_\ell}(\cdot)$.

Recall that $\nu_\ell$ is distributed uniformly on some interval $[a, b] = [a_\ell, b_\ell]$. A known fact about characteristic function of uniform distribution is that

$$\psi(\lambda) = \frac{e^{i\lambda b} - e^{i\lambda a}}{i\lambda(b - a)}.$$

The rest of the proof is a simple but somewhat tedious integration. By Esseen inequality,

$$Q_X(x) \leq x \int_{-2\pi/x}^{2\pi/x} |\psi(\lambda)| \, d\lambda \tag{C.16}$$

$$= x \int_{-2\pi/x}^{2\pi/x} \left| \frac{e^{i\lambda b} - e^{i\lambda a}}{i\lambda(b - a)} \right| d\lambda \tag{C.17}$$

$$= x \int_{-2\pi/x}^{2\pi/x} \frac{\sqrt{2 - 2\cos(\lambda b - \lambda a)}}{|\lambda(b - a)|} d\lambda \tag{C.18}$$

Let $u = \lambda(b-a)$. Then by substitution we have:

$$Q_X(x) \le x \int_{-2\pi(b-a)/x}^{2\pi(b-a)/x} \sqrt{\frac{2 - 2\cos(u)}{u^2}}\, du \tag{C.19}$$

$$= x \int_{-2\pi(b-a)/x}^{2\pi(b-a)/x} \sqrt{\frac{2 - 2\sum_{n=0}^{\infty} \frac{(-1)^n u^{2n}}{(2n)!}}{u^2}}\, du \qquad \text{(by Maclaurin series of } \cos(u))$$

$$= x \int_{-2\pi(b-a)/x}^{2\pi(b-a)/x} \sqrt{2}\sqrt{\frac{1}{2} - \frac{u^2}{4!} + \cdots}\, du \tag{C.20}$$

When we have $|u| \le 1$, the terms in the integrand are decreasing. Hence, the entire integrand can be upper bounded by 1. Otherwise, when $|u| \ge 1$, we can upper bound the integrand by $2/|u|$. Hence, the concentration function $Q_X(t)$ is upper bounded by:

$$Q_X(x) \le x \int_{-2\pi(b-a)/x}^{-1} \frac{2}{|u|}\, du + \int_{-1}^{1} 1\, du + \int_{1}^{2\pi(b-a)/x} \frac{2}{|u|}\, du$$

$$= x \int_{1}^{2\pi(b-a)/x} \frac{-2}{|v|}\, dv + \int_{-1}^{1} 1\, du + \int_{1}^{2\pi(b-a)/x} \frac{2}{|u|}\, du \qquad \text{(substitution } v = -u)$$

$$= x \left( -2\log(v) \Big|_{1}^{2\pi(b-a)/x} + 2 + 2\log(u) \Big|_{1}^{2\pi(b-a)/x} \right)$$

$$= 2x$$

For $x = \delta/2\cdot 8^d$, we have $Q_X\left(\delta/2\cdot 8^d\right) \le \delta/8^d$. Observe that since $A$ and $A'$ are subsets of atoms, there are at most $2^d$ possible choices for each arm $A$ and $A'$. Hence, for a fixed vector $n$, there are $4^d$ possible values of $X$. By union bound, we have $\mathbb{P}[\tau_{\mathcal{P}}(n) \le \delta/2\cdot 8^d] \le \delta/2^d$. $\qquad \square$

# D  Probabilistic tools from prior work

In this appendix, we spell out some probabilistic tools from prior work that we rely on.

### Bayesian Chernoff Bound

We use an easy corollary of the Bayesian Chernoff Bound from Sellke and Slivkins (2021).

**Lemma D.1** (Sellke and Slivkins (2021))**.** *Fix round $t$ and parameters $\epsilon, \tau > 0$. Suppose algorithm's history $\mathcal{F}_t$ almost surely contains at least $\epsilon^{-2}$ samples of each atom. Let $\tilde{\theta}$ be a posterior sample for the mean reward $\theta$, i.e., $\tilde{\theta}$ is an independent sample from the posterior distribution on $\theta$ given $\mathcal{F}_t$. Then for some universal absolute constant $C$, we have*

$$\mathbb{P}\left[ \left\| \tilde{\theta} - \theta \right\| \ge \tau\epsilon \right] \le C \cdot e^{-\tau^2/C}, \tag{D.1}$$

$$\mathbb{P}\left[ \left\| \mathbb{E}[\theta | \mathcal{F}_t] - \theta \right\| \ge \tau\epsilon \right] \le C \cdot e^{-\tau^2/C}. \tag{D.2}$$

*Proof.* Sellke and Slivkins (2021) contains this result for $d = 1$ atoms. Here, we apply the result from Sellke and Slivkins (2021) to each atom separately, using the fact that the Bayesian update is independent across atoms. $\qquad \square$

### Harris Inequality

We invoke Harris Inequality about correlated random variables.

**Theorem D.2** (Harris (1960))**.** *Let $f, g : \mathbb{R}^n \to \mathbb{R}$ be nondecreasing functions. Let $X_1, \cdots, X_n$ be independent real-valued random variables and define the random vector $X = (X_1, \cdots, X_n)$ taking values in $\mathbb{R}^n$. Then*

$$\mathbb{E}\left[ f(X)\, g(X) \right] \ge \mathbb{E}[f(X)]\, \mathbb{E}[g(X)]$$

*Similarly, if $f$ is nonincreasing and $g$ is nondecreasing then*

$$\mathbb{E}\,[\,f(X)\,g(X)\,] \leq \mathbb{E}[f(X)]\,\mathbb{E}[g(X)]$$

**Remark D.3** (Mixed-monotonicity Harris inequality)**.** If $f$ and $g$ are both increasing or both decreasing in each coordinate, then the results of Theorem D.2 still hold since we can simply negate some coordinates in the parameterization, i.e. we view $f$ and $g$ as increasing function of $-x_i$ instead of decreasing function of $x_i$. We refer to this in the proof as the mixed-monotonicity Harris inequality to highlight this subtle modification.

### Tails of sub-Gaussian distribution

**Lemma D.4.** *If random variable $X$ is $O(1)$-sub-Gaussian and event $E$ has probability $\mathbb{P}[E] \leq p$, then $\mathbb{E}[|X \cdot 1_E|] \leq O(p\sqrt{\log(1/p)})$*

## E   Initial exploration: reduction to Incentivized RL (proof of Theorem 4.6)

This appendix spells out the analysis for Section 4.2: the approach for initial exploration by reduction to incentivized reinforcement learning (RL) (Simchowitz and Slivkins, 2021).

We build on an algorithm from Simchowitz and Slivkins (2021), called HiddenHallucination, and their guarantee for this algorithm. We state their setup and guarantee below. (The specification of their algorithm is unimportant for our presentation.) Then we use it to prove Theorem 4.6.

### Incentivized RL: the setup and the guarantee

The setting is as follows. Consider an MDP with $S$ states, $A$ actions and $H$ stages, where $H$ is the time horizon. We write $x \in [S]$, $a \in [A]$, and $h \in [H]$ to represent states, actions, and stages, respectively. In the following analysis, we often refer to $(x, a, h)$ triples. We consider a set of feasible $(x, a, h)$ triples called FEASIBLE $\subset [S] \times [A] \times [H]$. (In Simchowitz and Slivkins (2021), FEASIBLE $= [S] \times [A] \times [H]$, but we will extend their result to an arbitrary FEASIBLE.)

The "true" MDP is denoted by $\phi$. Let $r_\phi(x, a, h)$ be the expected reward if action $a$ is chosen at state $x$ and stage $h$. We posit a Bayesian model: $\phi$ is chosen from a Bayesian prior $\mathcal{P}$.

Then, we consider the setting of *episodic RL*, where in each episode $t$ an algorithm chooses a policy $\pi^{(t)}$ in this MDP.[6] The chosen policies must satisfy a similar BIC condition: for each round $t \in [T]$,

$$\mathbb{E}[\,V(\pi) - \mu(\pi') \mid \pi^{(t)} = \pi\,] \geq 0 \qquad \forall \text{ policies } \pi, \pi' \in \mathcal{A} \text{ with } \mathbb{P}[\pi^{(t)} = \pi] > 0,$$

where $V(\pi)$ is the value (expected reward) of policy $\pi$. Essentially, this is the same condition as (1), where arms are replaced with policies.

We only need the guarantee for HiddenHallucination for an MDP with deterministic transitions (but randomized rewards). This guarantee depends on the following prior-dependent quantities:

$$q_{\mathrm{pun}}(\epsilon) \coloneqq \mathbb{P}[r_\phi(x, a, h) \leq \epsilon, \forall (x, a, h) \in \texttt{FEASIBLE}], \tag{E.1}$$

$$r_{\mathrm{alt}} \coloneqq \min_{(x,a,h) \in \texttt{FEASIBLE}} \mathbb{E}[r_\phi(x, a, h)]. \tag{E.2}$$

The guarantee is stated as follows.

**Theorem E.1.** *Consider an arbitrary prior $\mathcal{P}$. Fix parameters $\delta \in (0, 1]$. Assume that $r_{\mathrm{alt}} > 0$ and $q_{\mathrm{pun}} = q_{\mathrm{pun}}(\epsilon_{\mathrm{pun}}) > 0$, where $\epsilon_{\mathrm{pun}} = r_{\mathrm{alt}}/18H$. Consider HiddenHallucination with punishment parameter $\epsilon_{\mathrm{pun}}$, appropriately chosen phase length $n_{\mathrm{ph}}$, and large enough target $n = n_{\mathrm{lrn}}$. This algorithm is guaranteed to explore all $(x, a, h) \in \texttt{FEASIBLE}$ with probability at least $1 - \delta$ by round $N_0$, where $n$ and $N_0$ are specified below.*

*For some absolute constants $c_1, c_2$, it suffices to take*

$$n = n_{\mathrm{lrn}} \geq c_1 \cdot r_{\mathrm{alt}}^{-2} H^4 \left( S + \log \frac{SAH}{\delta \cdot r_{\mathrm{alt}} \cdot q_{\mathrm{pun}}} \right),$$

$$N_0 = c_2 \cdot n \cdot q_{\mathrm{pun}} \cdot r_{\mathrm{alt}}^{-3} \cdot SAH^4$$

---

[6]By default, MDP policies are Markovian and deterministic.

*In particular, for any $n \geq 1$, one can obtain $N_0 \leq n \cdot q_{\text{pun}} \cdot \text{poly}\left(r_{\text{alt}}^{-1} SAH\right) \cdot \log\left(\delta^{-1} q_{\text{pun}}^{-1}\right)$.*

As we mentioned above, Simchowitz and Slivkins (2021) guarantees Theorem E.1 for FEASIBLE $= [S] \times [A] \times [H]$. Below, we show how to extend it to an arbitrary FEASIBLE $= [S] \times [A] \times [H]$.

**Theorem E.2** (Simchowitz and Slivkins (2021))**.** *The guarantee in Theorem E.1 holds for* FEASIBLE $= [S] \times [A] \times [H]$.

**Remark E.3.** The relevant result, Theorem 5.5 in Simchowitz and Slivkins (2021) is stated for MDPs with randomized transitions. In this more general formulation, (E.1) and (E.2) are conditioned on an object called *censored ledger*, and then a minimum is taken over all such objects. However, this conditioning vanishes when the MDP transitions are deterministic. (This follows easily from Lemma 6.2 in Simchowitz and Slivkins (2021), essentially because censored ledgers do not carry any useful information.) We present a version without censored ledgers, because defining them is quite tedious.

*Proof Sketch for Theorem E.1.* Start with an arbitrary FEASIBLE. We modify the MDP as follows. Add two terminal state, GOOD and BAD, such that where we deterministically transition into BAD if $(x, a, h) \notin$ FEASIBLE. Otherwise, at the end of the MDP, we go into GOOD. We let BAD yield reward 0, and GOOD yield reward $H + 1$. With this modified MDP, even if all $(x, a, h)$ triples are allowed, any BIC algorithm would only choose feasible policies, *i.e.,* policies that only choose $(x, a, h) \in$ FEASIBLE. So, we conclude by invoking Theorem E.2. $\qquad\square$

### Proof of Theorem 4.6 for combinatorial semi-bandits

Now, let us go back to combinatorial semi-bandits and prove Theorem 4.6. We start with an instance of combinatorial semi-bandits and construct an MDP as specified in Section 4.2. Then we invoke Theorem E.1. To state the final guarantee, it remains to interpret (and simplify) the quantities in Theorem E.1 for a particular MDP obtained with our construction.

First, recall that $H = A = d$ and $S \leq d$, where $d$ is the number of atoms.

Second, $r_\phi(x, a, h)$ is simply $\theta_\ell$, the expected reward of the corresponding atom $\ell$. Accordingly,

$$r_{\text{alt}} = \min_{\text{atoms } \ell \in [d]} \mathbb{E}[\theta_\ell] \geq \min_{\text{priors } \mathcal{P}_\ell \in \mathcal{C}} \mathbb{E}_{\theta_\ell \sim \mathcal{P}_\ell}[\theta_\ell] := \epsilon_0.$$

Note that $\epsilon_0$ is determined by the collection $\mathcal{C}$ of feasible per-atom priors.

Finally, observe that $q_{\text{pun}}$ is the probability of all $(x, a, h)$ triples have low reward:

$$q_{\text{pun}}(\epsilon) = \mathbb{P}[\theta_\ell \leq \epsilon : \ \forall \text{ atoms } \ell \in [d]]$$

We can divide all $(x, h, a)$ triples into classes, where each class represents an atom. Since our prior is independent across the atoms,

$$
\begin{aligned}
q_{\text{pun}} = q_{\text{pun}}(r_{\text{alt}}) &\geq q_{\text{pun}}(\epsilon_0) \\
&= \mathbb{P}\left[\theta_\ell \leq \epsilon_0 : \ \forall \text{ atoms } \ell \in [d]\right] \\
&= \prod_{\ell \in [d]} \mathbb{P}\left[\theta_\ell \leq \epsilon_0\right] \\
&\geq \left(\min_{\text{priors } \mathcal{P}_\ell \in \mathcal{C}} \mathbb{P}_{\theta_\ell \sim \mathcal{P}_\ell}\left[\theta_\ell \leq \epsilon_0\right]\right)^d.
\end{aligned}
$$

Again, the expression in $(\cdot)$ is determined by the collection $\mathcal{C}$.