# OpenReview forum: "Incentivizing Combinatorial Bandit Exploration"
_NeurIPS.cc/2022/Conference — NeurIPS 2022 Accept_

### Official Review · Reviewer_gfpS · 2022-07-11

**Rating:** 7
**Confidence:** 3
**Soundness:** 4 excellent
**Presentation:** 3 good
**Contribution:** 3 good

**Summary:**

This work investigates incentivized exploration in a combinatorial semi-bandits setting. There are two main theoretical results. First, it was proven that Thompson sampling (TS), when concatenated with a first algorithm that generates at least given number of samples for each atom, is Bayesian incentive-compatible (BIC). Second, the authors propose two approaches, both adapted from the existing literature to handle the additional challenges of combinatorial semi-bandits, that satisfy the requirements of the first algorithm before TS.

**Questions:**

- The second approach (Sec. 4.2) seems an overkill given that Simchowitz and Slivkins (2021) handles the more difficult episodic RL setting. Is there any way to simplify?
- Line 291, should it be (11) instead of (7)?


**Ethics Review Area:**

["I don’t know"]

**Limitations:**

The authors did a great job explaining limitations and social impact in the Introduction section.

**Strengths And Weaknesses:**

Strengths:

1) The studied problem of incentivized exploration in combinatorial semi-bandits is novel. In particular, TS is considered as the bandit learning algorithm, which is both interesting and challenging.

2) Theoretically establishing that TS is BIC is a strong contribution.

3) The two approaches to satisfy the requirements of $\texttt{ALG}$ are interesting, particularly in the combinatorial semi-bandits setting.

Weaknesses:

- There lacks a discussion of motivating/matching applications of incentivized exploration (which is well motivated in prior papers) specifically for combinational semi-bandits. It was briefly mentioned in Page 2 but I feel that a detailed example would be beneficial.
- A discussion of how to handle $K>2$ in Section 3.1 would be valuable.
- It is worth discussing the constant (number of samples) in the theorems. Will they blow up $T$?

---

> ### Author Response · Authors · 2022-08-01
> **Response to Reviewer gfpS**
>
> We thank the reviewer for their review. Please see our responses below:
>
> Re motivation: our primary motivation is to extend the learning problem in incentivized exploration (see the response to reviewers CaZr and jo2d). Aside from this abstract motivation, we list several stylized motivating examples on page 2, as the reviewer has pointed out. We frame each example as a recommendation system that recommends "actions" with a combinatorial structure. A user then can choose to follow such recommendation (i.e., pursue the action), or deviate from it. The BIC condition ensures the former, in which case the user reports a "reward" for each atom in the action. Thus:
>
> 1. Recommending online content such as news/entertainment articles. Here, an "action" is a slate of articles. The platform recommends each user a slate of articles, and the user can choose to "follow the recommendation": look at each article in the slate. The feedback for each article is either generated automatically from the interaction (e.g., dwell time or how far the user has scrolled) or entered explicitly (e.g., "thumbs up" or "thumbs down").
>
> 2. Recommending complementary products: e.g., a three-piece suit or the contents of a child's pencil box. Thus, an "action" is a slate of products. A user who follows such a recommendation buys each item in the slate, and eventually reports its quality (e.g., "thumbs up" or "thumbs down").
>
> 3. Recommending driving directions. Here, an "action" consists of a route, which is a sequence of segments. A driver can choose to follow this route, in which case the travel time for each segment is reported (e.g., automatically by an app).
>
> Our framing differs from the "usual" framing for combinatorial semi-bandits, much like the framing for incentivized exploration differs from that for bandits. We'll make this point more explicit in the revision.
>
> Re handling $K>2$ in Section 3.1: Our main result (Theorem 3.1) relies on a structural assumption: independence across atoms, which is very natural for combinatorial semi-bandits. The result for $K=2$ in Section 3.1 is just a "tangent": the basic case when Thompson Sampling is BIC under _arbitrary_ correlation across arms. We can't handle $K>2$ arms without structural assumptions.
>
> Re the constants in the theorems: we analyze a sufficient number of samples (resp., rounds) in all our results: see Corollaries 3.4, 4.3, and 4.5 and Theorem 4.6. These numbers are determined by the prior and do not depend on T.
>
> Re "The second approach seems an overkill": a non-trivial "translation" from the MDP result (Simchowitz & Slivkins, 2021) provided in Section 4.2 suffices for our purposes. We agree that their algorithm and analysis simplify considerably when specialized to combinatorial semi-bandits. However, we think such specialization would still be very technical (e.g., the simplifications would be mainly to the material in their appendices), and would not improve the asymptotic guarantee in our Theorem 4.6.
>
> Re typo in line 291: Thanks for noticing! We've addressed this typo in Appendix A (Errata).
>
> We hope our responses clarify any questions the reviewer may have had about our paper.

---

> > ### Comment · Reviewer_gfpS · 2022-08-05
> > **Follow-up questions**
> >
> > Thanks for the clarification! Regarding $K>2$, I was hoping to see some discussion on an arbitrary correlation matrix among the atoms. Is this possible? Regarding the constants in theorems, for example, the definition of $n_{\text{ts}}$ in Theorem 3.1 would naturally raise the question: what if this required number of samples is bigger than $T$?

---

> > > ### Author Response · Authors · 2022-08-05
> > > **Response to Reviewer gfpS follow-up questions**
> > >
> > > Thanks for following up!
> > >
> > > Re: arbitrary correlation matrix across atoms
> > >
> > > We need independence to apply Harris inequality. But this is a great question! We discussed replacing Harris inequality with FKG inequality, which admits weaker assumptions. We succeeded in pushing these assumptions through our analysis, but in the end, we needed a rather strange-looking assumption on the prior which we haven’t been able to motivate. So we decided that this extension would only confuse the reader. But we can include it in an appendix when we revise.
> > >
> > > On the other hand, we conjecture that the BIC property would not hold with unlimited correlation. But we can't prove it. In fact, no such results are known for _any_ reasonable bandit algorithm, not just Thompson Sampling (*). We'll be sure to elaborate on this in the revision.
> > >
> > > (*) There are no known theorems of the form "for some non-degenerate priors, this algorithm is not BIC for any prior-determined amount of initial data". (But there are "blanket" results of the form: for these degenerate priors, no algorithm is BIC whatsoever.)
> > >
> > > Re: what if #samples $>T$ in Theorem 3.1
> > >
> > > Let ALG be the algorithm that collects the initial samples, and suppose it completes in $T_0$ rounds, where $T_0$ is determined by the prior. Our theorem asserts that Thompson Sampling will be BIC afterward. So, if we require ALG to be BIC, and we have $T$ rounds total, then we need $T>T_0$. (Note that $T_0$ is a "constant" as far as $T$ is concerned.) However, the theorem admits an alternative interpretation in which the initial samples are provided exogenously (e.g., bought) before Thompson Sampling starts (see Remark 3.5).

---

> > > > ### Comment · Reviewer_gfpS · 2022-08-08
> > > > **Thank you for the response**
> > > >
> > > > I have no further questions.

---

### Official Review · Reviewer_tHsD · 2022-07-13

**Rating:** 3
**Confidence:** 2
**Soundness:** 2 fair
**Presentation:** 2 fair
**Contribution:** 1 poor

**Summary:**

This paper studied incentivized exploration problem, and provided some theoretic study of using combinatorial semi-bandits to solve the problem. The paper does not seem to be complete, since there is no experimental sections to show the results to support the proposed algorithm.

**Questions:**

Incentivizing Combinatorial Bandit is an interesting research topic. However, without experiment justification, there is no point to show the paper in the high standard NeurIPS conference.

**Limitations:**

Since there is no experiment result presented in the paper, it is hard to reproduce the work.

**Strengths And Weaknesses:**

The mathematical framework seems ok, including Thompson Sampling, BIC algorithm and so on. However, there is no experimental justification at all.

---

> ### Author Response · Authors · 2022-08-01
> **Response to Reviewer tHsD**
>
> We thank the reviewer for their review. Regarding the lack of experiments:
>
> The reviewer appears to make a very general claim that theoretical papers without experiments are not suitable for NeurIPS. Empirically, this is not the case: many such papers have been published in NeurIPS over the years. Our own experience as neurips (meta-)reviewers is that NeurIPS is open to theory-only contributions, particularly for the intersection of ML and economic theory.
>
> Furthermore, we think the experimental evaluation is not particularly appropriate for our paper, for the following two reasons:
>
> 1. Our main result concerns the economic properties of Thompson Sampling, without any modification to the original algorithm. Thompson Sampling is a standard algorithm whose empirical performance is fairly well-studied in a variety of settings. We think that further empirical evaluation of Thompson Sampling is not very relevant to the study of incentivized exploration.
>
> 2. The performance of "initial sampling" is guaranteed within the claimed number of rounds that is determined by the prior. So, there is nothing to evaluate, aside from the number of rounds itself which is just a formula. Besides, this number is only meant as a theoretical proof of concept for the claimed dependence on the number of atoms $d$, and is not optimized for practice (essentially because the same holds for the two techniques from prior work that we build on).

---

### Official Review · Reviewer_jo2d · 2022-07-14

**Rating:** 6
**Confidence:** 3
**Soundness:** 3 good
**Presentation:** 3 good
**Contribution:** 3 good

**Summary:**

This paper investigates the incentivized exploration problem in combinatorial semi-bandits setting. The authors prove that Thompson Sampling is incentive-compatible when initialized with a sufficient number of samples of each arm in combinatorial semi-bandits. In addition, the authors design

**Questions:**

N/A

**Strengths And Weaknesses:**

The paper is well-written and compares with previous literature thoroughly. The main results of this paper are solid, however, I have no enough expertise to judge whether the proof technique is trivial or not compared with the existing literature. The initial exploration algorithms seems rely on the reduction to previous literature and the BIC-bandit expert can help to judge the technical contribution of this paper.

Weaknesses: This paper is a bit incremental from my perspective, which generalizes previous incentivized exploration to combinatorial semi-bandit problems. The significance of this paper needs to be justified further by BIC-bandit expert.

Overall, this is a good theoretical paper. Whether to accept the paper depends on the technical contribution of this paper.

---

> ### Author Response · Authors · 2022-08-01
> **Response to Reviewer jo2d**
>
> We thank the reviewer for their review. As the questions raised by reviewer jo2d are similar to that of reviewer CaZr, we opt to use the same response for both reviews. Regarding the novelty and significance of our results:
>
> The basic model of incentivized exploration is very stylized and can be extended along two "dimensions": more sophisticated economic models for agents' behavior and incentives, and more complex machine-learning models for actions' structures and rewards. All published work has only dealt with the former, whereas we pursue the latter. In particular, all published work focuses on small, unstructured action sets and independent priors (i.e., priors that are independent across arms). In contrast, we venture into large, structured action sets with highly correlated priors (as we point out in L24-29). We consider combinatorial semi-bandits as a paradigmatic exploration problem with large, structured action sets, and model correlated priors via independence across atoms.
>
> Our main contribution is proving that Thompson Sampling is BIC for combinatorial semi-bandits. While Selke & Slivkins (2021) has shown a similar result for "classical" multi-armed bandits, their assumption of independent priors does not hold in the highly-correlated structure of combinatorial bandits. Hence, we analyze the incentive-compatibility condition at the atom level in order to use the main technical tool of mixed-monotonicity Harris inequality.
>
> For the initial exploration, our main result (in Section 4.1) builds a substantial "super-structure" on top of the "hidden exploration" paradigm from Mansour et al. (2020). We provide a _new intepretation_ of their guarantee in terms of a generic sequence of arms that satisfies a certain property (Theorem 4.1). Then, we use the structure of combinatorial semi-bandits to construct a suitable sequence of arms for which this property holds; this is the main technical contribution in this result.
>
> Our alternative approach for initial exploration (with a somewhat worse dependence on the number of atoms) provides a new and non-trivial "translation" from a similar result on incentivized reinforcement learning (which comes from the yet-unpublished working paper of Simchowitz and Slivkins (2021)).

---

### Official Review · Reviewer_CaZr · 2022-07-18

**Rating:** 5
**Confidence:** 2
**Soundness:** 3 good
**Presentation:** 3 good
**Contribution:** 2 fair

**Summary:**

This paper considers a combinatorial multi-armed bandits (CMAB) setting where multiple users interact with a learning algorithm. The algorithm recommends arms and the users decide whether they want to follow algorithm's recommendation (explore) or choose different arms (exploit). The paper considers the problem of incentivizing identical Bayesian users to follow the algorithm's recommendation. Precisely, an algorithm is considered Bayesian Incentive Compatible (BIC) if for every trial the algorithm plays an arm that has the best expected reward with respect to prior belief of a given user. The paper shows that the Thompson sampling algorithm for CMAB satisfies BIC given a sufficient number of bootstrap samples for each arm. The paper then considers the goal of designing BIC algorithms for providing bootstrap samples and proposes two algorithms: (1) based on a reduction to incentivized K-armed bandits problem, (2) based on a reduction to incentivized reinforcement learning.

**Questions:**

I would request the authors to specify the challenges in achieving incentivized exploration for combinatorial bandits, and the technical ideas used to overcome these challenges.

**Limitations:**

I do not foresee any negative societal impact.

**Strengths And Weaknesses:**

Strengths: The paper is very well-written with proper intuition for various assumptions/ideas. Prior work has also been addressed adequately.

Weaknesses: I think that the contribution of the paper is incremental in nature given the amount of existing work on incentivizing exploration. For example, it was already shown by Selke & Slivkins (2021) that Thompson sampling is BIC for classical bandits and it does not seem surprising that the same result also holds for the combinatorial bandits setting (albeit for a different number of bootstrap samples). The bootstrap algorithms proposed by the paper are also based on existing ideas in Mansour et al. (2020) and Simchowitz & Slivkins (2021).

---

> ### Author Response · Authors · 2022-08-01
> **Response to Reviewer CaZr**
>
>
> We thank the reviewer for their review. Regarding the novelty and significance of our results:
>
> The basic model of incentivized exploration is very stylized and can be extended along two "dimensions": more sophisticated economic models for agents' behavior and incentives, and more complex machine-learning models for actions' structures and rewards. All published work has only dealt with the former, whereas we pursue the latter. In particular, all published work focuses on small, unstructured action sets and independent priors (i.e., priors that are independent across arms). In contrast, we venture into large, structured action sets with highly correlated priors (as we point out in L24-29). We consider combinatorial semi-bandits as a paradigmatic exploration problem with large, structured action sets, and model correlated priors via independence across atoms.
>
> Our main contribution is proving that Thompson Sampling is BIC for combinatorial semi-bandits. While Selke & Slivkins (2021) has shown a similar result for "classical" multi-armed bandits, their assumption of independent priors does not hold in the highly-correlated structure of combinatorial bandits. Hence, we analyze the incentive-compatibility condition at the atom level in order to use the main technical tool of mixed-monotonicity Harris inequality.
>
> For the initial exploration, our main result (in Section 4.1) builds a substantial "super-structure" on top of the "hidden exploration" paradigm from Mansour et al. (2020). We provide a _new intepretation_ of their guarantee in terms of a generic sequence of arms that satisfies a certain property (Theorem 4.1). Then, we use the structure of combinatorial semi-bandits to construct a suitable sequence of arms for which this property holds; this is the main technical contribution in this result.
>
> Our alternative approach for initial exploration (with a somewhat worse dependence on the number of atoms) provides a new and non-trivial "translation" from a similar result on incentivized reinforcement learning (which comes from the yet-unpublished working paper of Simchowitz and Slivkins (2021)).

---

### Meta-Review · Area_Chair_LNxZ · 2022-08-23

**Recommendation:** Accept
**Confidence:** Certain

**Metareview:**

This paper investigates the problem of incentivized exploration in the combinatorial semi-bandits setting.  The reviewers are overall positive about the paper. The main concern of the paper is that the contribution is incremental given the number of prior works on incentivizing exploration.  The authors' responses have more explicitly addressed this concern, and we encourage the authors to incorporate their responses into the paper.  There have also been issues brought up about the lack of experiments, but I agree with the authors that given the main contribution of this work is theoretical, having empirical evaluations is a plus but not required.  Overall, I believe the contribution of the paper outweighs the concerns and would therefore recommend acceptance.

**Award:**

No

---

### Decision · Program_Chairs · 2022-09-14

Accept